# Medical Dead-ends and Learning to Identify High-risk States and Treatments

**Mehdi Fatemi**
Microsoft Research
mehdi.fatemi@microsoft.com

**Taylor W. Killian**
University of Toronto, Vector Institute
twkillian@cs.toronto.edu

**Jayakumar Subramanian**
Media and Data Science Research, Adobe India
jayakumar.subramanian@gmail.com

**Marzyeh Ghassemi**
Massachusetts Institute of Technology
mghassem@mit.edu

## Abstract

Machine learning has successfully framed many sequential decision making problems as either supervised prediction, or optimal decision-making policy identification via reinforcement learning. In data-constrained offline settings, both approaches may fail as they assume fully optimal behavior or rely on exploring alternatives that may not exist. We introduce an inherently different approach that identifies possible "dead-ends" of a state space. We focus on the condition of patients in the intensive care unit, where a "medical dead-end" indicates that a patient will expire, regardless of all potential future treatment sequences. We postulate "treatment security" as avoiding treatments with probability proportional to their chance of leading to dead-ends, present a formal proof, and frame discovery as an RL problem. We then train three independent deep neural models for automated state construction, dead-end discovery and confirmation. Our empirical results discover that dead-ends exist in real clinical data among septic patients, and further reveal gaps between secure treatments and those that were administered.

## 1 Introduction

Off-policy Reinforcement Learning (RL) was designed as the way to isolate behavioural policies, which generate experience, from the target policy, which aims for optimality. It also enables learning multiple target policies with different goals from the same data-stream or from previously recorded experience [1]. This algorithmic approach is of particular importance in safety-critical domains such as robotics [2], education [3] or healthcare [4] where data collection should be regulated as it is expensive or carries significant risk. Despite significant advances made possible by off-policy RL combined with deep neural networks [5–7], the performance of these algorithms degrade drastically in fully *offline* settings [8], without additional interactions with the environment [9, 10]. These challenges are deeply amplified when the dataset is limited and exploratory new data cannot be collected for ethical or safety purposes. This is because robust identification of an optimal policy requires exhaustive trial and error of various courses of actions [11, 12]. In such fully offline cases, naively learned policies may significantly overfit to data-collection artifacts [13–15]. Estimation errors due to limited data may further lead to mistimed or inappropriate decisions with adverse safety consequences [16].

Even if optimality is not attainable in such constrained cases, negative outcomes in data can be used to identify behaviors to avoid, thereby guarding against overoptimistic decisions in safety-critical domains that may be significantly biased due to reduced data availability. In one such domain, healthcare, RL has been used to identify optimal treatment policies based on observed outcomes of

past treatments [17]. These policies correspond to advising *what treatments to administer*, given a patient's condition. Unfortunately, exploration of potential courses of treatment is not possible in most clinical settings due to legal and ethical implications; hence, RL estimates of optimal policies are largely unreliable in healthcare [18].

In this paper, we develop a novel RL-based method, Dead-end Discovery (DeD), to identify *treatments to avoid* as opposed to what treatment to select. Our paradigm shift avoids pitfalls that may arise from constraining policies to remain close to possibly suboptimal recorded behavior as is typical in current state of the art offline RL approaches [10, 19–21]. When the data lacks sufficient amounts of exploratory behavior, these methods fail to attain a reliable policy. We instead use this data to constrain the scope of the policy, based on retrospective analysis of observed outcomes, a more tractable approach when data is limited. Our goal is *to avoid future dead-ends* or regions in the state space from which negative outcomes are inevitable (formally defined in Section 3.2). DeD identifies dead-ends via two complementary Markov Decision Processes (MDPs) with a specific reward design so that the underlying value functions will carry special meaning (Section 3.4). These value functions are independently estimated using Deep Q-Networks (DQN) [5] to infer the likelihood of a negative outcome occurring (D-Network) and the reachability of a positive outcome (R-Network). Altogether DeD formally connects the notion of value functions to the dead-end problem, learned directly from offline data.

We validate DeD in a carefully constructed toy domain, and then evaluate real health records of septic patients in an intensive care unit (ICU) setting [22]. Sepsis treatment and onset is a common task in medical RL [23–26] because the condition is highly prevalent [27, 28], physiologically severe [29], costly [30] and poorly understood [31]. Notably, the treatment of sepsis itself may also contribute to a patient's deterioration [32, 33], thus making treatment avoidance a particularly well-suited objective. We find that DeD confirms the existence of dead-ends and demonstrate that 12% of treatments administered to terminally ill patients reduce their chances of survival, some occurring as early as 24 hours prior to death. The estimated value functions underlying DeD are able to capture significant deterioration in patient health 4 to 8 hours ahead of observed clinical interventions, and that higher-risk treatments possibly account for this delay. Early identification of suboptimal treatment options is of great importance since sepsis treatment has shown multiple interventions within tight time frames (10 to 180 minutes) after suspected onset decreases sepsis mortality [34].

While motivated by healthcare, we propose the use of DeD in safety-critical applications of RL in most data-constrained settings. We introduce a formal methodology that outlines how DeD can be implemented within an RL framework for use with real-world offline data. We construct and train DeD in a generic manner which can readily be used for other data-constrained sequential decision-making problems. In particular, we emphasize that DeD is well suited to analyze high-risk decisions in real-world domains.

## 2 Related Work

**RL in Health:** RL has been the subject of much focus in health [17], with particular emphasis on sepsis seeking to develop optimal treatment recommendation policies [23–26, 35–38]. However, with fixed retrospective medical data, an optimal policy that maximizes a patient's chance of recovery is both computationally and experimentally infeasible. To our knowledge, we are the first to target improved treatment recommendations by avoiding high-risk treatments in a fully offline manner.

**Safety in RL:** RL has a rich history in safety [39], with recent work attempting to limit high risk actions by constraining parametric uncertainty [40], through alignment between agent and human objectives [41, 42], by directly constraining the agent optimization process to avoid unsafe actions [43], or by improving over a baseline policy [44]. In these settings model performance is evaluated in online settings where more data can be acquired or models can be tested against new cases as well as known baselines. We focus on the more challenging offline setting with limited and non-exploratory data, reflecting the reality of healthcare settings.

**Dead-ends:** The concept of dead-ends and the corresponding security condition that we build from was proposed by Fatemi et al. [45] in the context of *exploration*. In their work an online RL agent needs to experience various courses of actions from each state, through which it learns optimal behavior. We adapt this approach and expand the theoretical results to an offline RL setting as is found within healthcare–where exploration is untenable–to determine which treatments increase the likelihood of entering a dead-end, based on the patient's current health state.

Related concepts to dead-ends were introduced by Irpan et al. [46], focused primarily on policy evaluation. The authors introduce a notion of *feasible* states as those that are not *catastrophic* and from which an agent will not immediately fail. Whether or not a state is feasible is determined via positive-unlabeled classification. This inherently differs from our approach where we formally characterize dead-ends and a corresponding security condition through which we can identify treatments to avoid that likely lead to dead-ends[1]. Our formalization is discussed in the next section.

## 3   Methods

### 3.1   Preliminaries

Our pipeline isolates state construction from value estimation with RL. Therefore we consider episodic Markov Decision Processes (MDP) $\mathcal{M} = (\mathcal{S}, \mathcal{A}, T, r, \gamma)$, where $\mathcal{S}$ and $\mathcal{A}$ are the discrete sets of states and treatments[2]; $T : \mathcal{S} \times \mathcal{A} \times \mathcal{S} \to [0, 1]$ is a function that defines the probability of transitioning from state $s_t$ to $s_{t+1}$ if treatment $a_t$ is administered; $R : \mathcal{S} \times \mathcal{A} \times \mathcal{S} \to [r_{min}, r_{max}]$ is a finite reward function and $\gamma \in [0, 1]$ denotes a scalar discount factor.

A *policy* $\pi(s, a) = \mathbb{P}[A_t = a | S_t = s]$ defines how treatments are selected, given a state. A *trajectory* is comprised of sequences of tuples $(S_t, A_t, R_t, S_{t+1})$ with $S_0$ being the initial state of the trajectory. Sequential application of the policy is used to construct trajectories. The reward collected over the course of a trajectory induces the *return* $G_t = \sum_{j=0}^{\infty} \gamma^j R_{t+j+1}$. We assume that all the returns are finite and bounded. A trajectory is considered *optimal* if its return is maximized. A state-treatment value function $Q^\pi(s, a) = \mathbb{E}^\pi[G_0 | S_0 = s, A_0 = a]$ is defined in conjunction with a policy $\pi$ to evaluate the expected return of administering treatment $a$ at state $s$ and following $\pi$ thereafter. The optimal state-treatment value function is defined as $Q^*(s, a) = \max_\pi Q^\pi(s, a)$, which is the maximum expected return of all trajectories starting from $(s, a)$. We define state value and optimal state value as $V^\pi(s) = \mathbb{E}_{a \sim \pi} Q^\pi(s, a)$ and $V^*(s) = \max_a Q^*(s, a)$.

### 3.2   Special States

We define a terminal state as the final observation of any recorded trajectory. We focus on two types of terminal state that correspond to positive or negative outcomes. Our goal is to identify all *dead-end* states, from which negative outcomes are unavoidable (happening w.p.1), regardless of future treatments. In safety-critical domains, it is crucial to avoid such states *and* identify the probability with which any available treatment will lead to a dead-end. We also introduce the complementary concept of *rescue* states, from which a positive outcome is *reachable* with probability one. If an agent is in a rescue state, there exists at least one treatment at each time step afterwards which leads to either another rescue state or the eventual positive outcome. The fundamental contrast between dead-end and rescue states is that if the agent enters a rescue state, it *does not* mean the treatment process is done; it rather means that at each time step afterwards there exists at least one treatment to be found and administered until the positive outcome occurs. There might be trajectories starting from a rescue state which include non-rescue states. This is not the case for a dead-end state.

Formally, we augment $\mathcal{M}$ with a non-empty termination set $\mathcal{S}_T \subset \mathcal{S}$, which is the set of all terminal states. Mathematically, a terminal state is absorbing (self-transition w.p.1) with zero reward afterwards. All terminal states are by definition zero-valued, but the transitions to them may be associated with a non-zero reward. We require that, from all states, there exists at least one trajectory with non-zero probability arriving at a terminal state. In an offline setting with limited and non-exploratory data, inducing an optimal policy *is not feasible* [12]; hence, we do not specify the reward function of $\mathcal{M}$ for which standard RL would optimize cumulative rewards, but in later sections present a specific design of reward (and discount factor) to assist in identifying dead-end/rescue states. Finally, the sets of dead-end and rescue states are denoted respectively by $\mathcal{S}_D$ and $\mathcal{S}_R$. We formally distinguish dead-end/rescue states from the outcome, asserting that $\mathcal{S}_D, \mathcal{S}_R \not\subset \mathcal{S}_T$.

---

[1]A more in depth discussion on the differences between Irpan et al. [46] and this work can be found in Appendix Section A2

[2]Our results can easily be extended to continuous state-spaces by properly replacing summations with integrals. For brevity, we only present formal proofs for the discrete case. Additionally, as our primary motivating domain lies within healthcare, we use the term "treatment" in place of "action".

## 3.3 Treatment Security

When dealing with data-constrained offline scenarios, a core distinction is necessary: Realization of an optimal treatment at a given state requires knowledge of all future outcomes for all possible treatments, which is not feasible. However, the data may contain enough information to estimate the possible outcome of a certain treatment at a similar state. If such an outcome is negative with high probability, then we should advise against the treatment, even if an optimal treatment still remains unknown. This distinction leads to a paradigm shift from finding the best possible treatment to mindful avoidance of dangerous ones. This shift further motivates a different design space to make use of the limited, yet available data.

We adapt the security condition from Fatemi et al. [45] and formalize the treatment avoidance problem with a more generalized *treatment security condition*. We note that the chance of a negative outcome is best described by the probability of falling into a dead-end or immediate negative termination. The security condition therefore constrains the scope of a given behavioral policy $\pi$ if *any* knowledge exists about dead-ends or negative termination. Formally, if at state $s$, treatment $a$ leads to a dead-end with probability $P_D(s,a)$ or immediate negative termination with probability $F_D(s,a)$ with a level of certainty $\lambda \in [0,1]$, then $\pi$ must avoid selecting $a$ at $s$ with the same certainty:

$$P_D(s,a) + F_D(s,a) \geq \lambda \implies \pi(s,a) \leq 1 - \lambda. \tag{1}$$

E.g., if a treatment leads to a dead-end or termination with probability more than $80\%$, then that treatment should be selected for administration no more than $20\%$ of the time. While we would like (1) to hold for the maximum $\lambda$, inferring such maximal values is intractable for all state-treatment pairs. Moreover, directly computing $P_D$ and $F_D$ would require explicit knowledge of all dead-end and negative terminal states as well as all transition probabilities for future states. These make the application of (1) nearly impossible. We next develop a learning paradigm to enable (1) from data.

## 3.4 Dead-end Discovery (DeD)

In order to identify and confirm the existence of dead-end states, we construct two Markov Decision Processes (MDPs) $\mathcal{M}_D$ and $\mathcal{M}_R$ to be identical to $\mathcal{M}$, with $\gamma = 1$ for both. We also define the following reward functions: $\mathcal{M}_D$ returns $-1$ with any transition to a negative terminal state (and zero with all other transitions) whereas $\mathcal{M}_R$ returns $+1$ with any transition to a positive terminal state (zero otherwise). Let $Q_D^*, Q_R^*, V_D^*$ and $V_R^*$ denote the optimal state-treatment and state value functions of $\mathcal{M}_D$ and $\mathcal{M}_R$, respectively. Note that due to the reward functions of these MDPs, for all states and treatments, $Q_D^*(s,a) \in [-1, 0]$ and $Q_R^*(s,a) \in [0, 1]$.

Having selected treatment $a$ at state $s$, using the Bellman equation, we prove[3] that

$$-Q_D^*(s,a) = P_D(s,a) + F_D(s,a) + M_D(s,a) \tag{2}$$

In addition to the quantities defined previously, $M_D(s,a)$ denotes the probability of circumstances in stochastic environments where a negative terminal state ultimately occurs despite receiving optimal treatments at all steps in the future. Equation (2) therefore reveals that $-Q_D^*$ carries special physical meaning: it corresponds to the *minimum probability of a negative outcome*, because future treatments may not necessarily be optimal. Equivalently, $1 + Q_D^*(s,a)$ can be seen as the *maximum hope of a positive outcome* if treatment $a$ is administered at state $s$.

Building from Fatemi et al. [45], we show that $V_D^*$ of all dead-end states will be precisely $-1$. By extension, $Q_D^*(s,a) = -1$ for all treatments $a$ at state $s$ if and only if $s$ is a dead-end. In fact, $1 + Q_D^*(s,a)$ provides an appropriate threshold to secure any given policy $\pi(s,a)$. More formally, the following statement guarantees treatment security as presented in (1) for all values of $\lambda$:

$$\pi(s,a) \leq 1 + Q_D^*(s,a) \tag{3}$$

In short, for treatment security it is sufficient to abide by the maximum hope of a positive outcome. This construction directly connects the RL concept of value functions to dead-end discovery. While $V_D^*(s)$ enables detecting dead-end states, (3) leverages $Q_D^*$ for treatment avoidance. We establish parallel results for rescue states similarly. The following theorem summarizes the theory and shapes the basis of DeD. See Appendix A1 for the proof and further details.

---

[3]All proofs to the theoretical claims presented in this paper can be found in Appendix A1

**Theorem 1**. Let treatment $a$ be administered at state $s$, and $P_D(s, a)$ and $P_R(s, a)$ denote the probability of transitioning to a dead-end or rescue state. Similarly, let $F_D(s, a)$ and $F_R(s, a)$ denote the probability of transitioning to either a negative or positive terminal state. The following hold:

T1 $P_D(s, a) + F_D(s, a) = 1$ if and only if $Q_D^*(s, a) = -1$.

T2 $P_R(s, a) + F_R(s, a) = 1$ if and only if $Q_R^*(s, a) = 1$.

T3 There exists a threshold $\delta_D \in (-1,\ 0)$ independent of states and treatments, such that $Q_D^*(s, a) \geq \delta_D$ for all $s$ and $a$, unless $P_D(s, a) + F_D(s, a) = 1$.

T4 There exists a threshold $\delta_R \in (0,\ 1)$ independent of states and treatments, such that $Q_R^*(s, a) \leq \delta_R$ for all $s$ and $a$, unless $P_R(s, a) + F_R(s, a) = 1$.

T5 For any policy $\pi$, state $s$, and treatment $a$, if $\pi(s, a) \leq 1 + Q_D^*(s, a)$ and $\lambda \in [0, 1]$ exists such that $P_D(s, a) + F_D(s, a) \geq \lambda$, then $\pi(s, a) \leq 1 - \lambda$.

T6 For any policy $\pi$, state $s$, and treatment $a$, if $\pi(s, a) \geq Q_R^*(s, a)$ and $\lambda \in [0, 1]$ exists such that $P_R(s, a) + F_R(s, a) \geq \lambda$, then $\pi(s, a) \geq \lambda$.

It is immediate from (T1) and (T2) that $Q_D^*$ and $Q_R^*$ incorporate complete information when transitioning to a dead-end state or to a rescue state as a result of administrating treatment $a$ at $s$. (T3) assures that a threshold $\delta_D$ exists to separate treatments that lead immediately to dead-ends from alternatives. (T4) allows us to confirm a dead-end by examining if $Q_R^*$ is also smaller than some threshold $\delta_R$. No dead-end can violate $\delta_R$ due to (T4) and such a threshold exists. If $Q_D^*$ is available and $\delta_D$ is known, then this step is redundant. However, without access to $Q_D^*$ and an accurate $\delta_D$, (T4) helps to confirm any presumed dead-end. Finally, (T5) provides the means by which the treatment policy is guided to avoid dangerous treatments. (T6) is used to also confirm whether the treatment should be avoided. We explain how to practically select the thresholds $\delta_D$ and $\delta_R$ in Sec. 5.

Of note, by definition, value functions encompass long-term consequences and are not myopic to possible immediate events, as opposed to supervised learning from immediate observation of an outcome. This inherent characteristic of value functions indeed yields the theoretical result presented by Lemma 2 (Appendix Sec. A1), one result of which is that $-Q_D$ corresponds to the minimum probability of a negative future outcome. Supervised learning from immediate outcomes, on the other hand, lacks this formal property [47]; hence, it is not expected to provide parallel results with DeD.

### 3.5 Neural Network Based State Construction and Identification

**State construction (SC-Network).** In domains where solitary observations do not carry salient information for learning the decision-making process, states may need to be constructed from data using a neural network. In these circumstances a separate SC-Network can be used to transform a single or possible sequence of observations into a fixed embedding, considered the state $s$ at time $t$.

**Identification (D-Network and R-Network).** In order to approximate $Q_D^*$ and $Q_R^*$, two separate neural networks can be used to compute $Q_D$ and $Q_R$ for all treatments given a state constructed by the SC-Network. With trained $Q_D$ and $Q_R$ networks, we can then apply thresholds $\delta_D$ and $\delta_R$ as specified in Theorem 1. As data is limited and non-exploratory, approximation error is inevitable. To mitigate this limitation, the method's sensitivity can be adjusted by adapting the thresholds $\delta_D$ and $\delta_R$ (additionally, see Proposition 1 and Remarks 1-4 in Appendix A1). Smaller thresholds result in more false negative and less false positive cases. Of note, value-overestimation, a known limitation of deep RL models, will often cause $Q_D$ and $Q_R$ to be larger than $Q_D^*$ and $Q_R^*$ respectively. This naturally reduces false positives while increasing false negatives.

### 3.6 Toy Problem Validation: Life-Gate

We briefly provide a tabular toy-example (Life-Gate), which involves dead-ends, to empirically illustrates the merit of Theorem 1 by learning $Q_D^*$ and $Q_R^*$ (Figure 1). This toy set-up comprises an interesting case, where the agent faces an environment to examine with no knowledge of possible dangers. Importantly, once a dead-end state (yellow) is reached, it may take some random number of steps before reaching a "death gate" (red). All along such trajectories of dead-end states, the agent still has to choose actions with the (false) hope of reaching a "life-gate" (blue). Discovering any single dead-end state and signaling the agent when it is approached would be of significant

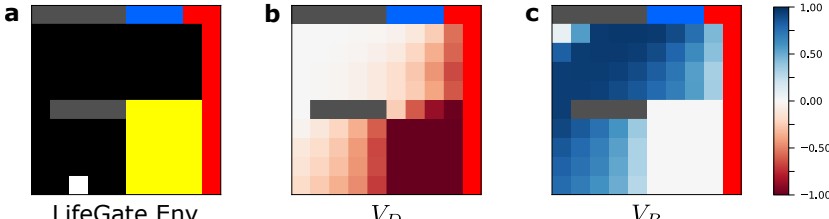

Figure 1: **The Life-Gate Example.** The tabular navigation task of life-gate is illustrated in (**a**). Corresponding dead-end and rescue state-value functions, $V_D$ and $V_R$, are shown in (**b**) and (**c**). The value functions are learned through Q-learning and with the definition of $\mathcal{M}_D$ and $\mathcal{M}_R$.

importance. On the other hand, adjacent states to dead-ends are possibly the most critical to alert, as it might be the last chance to still do something to avoid failure (see Appendix A3 for more details).

We next use the tools provided by Theorem 1. The value functions are more than 90% trained, still allowing learning errors. In this example, even with the errors due to lack of full convergence, $\delta_D = -0.7$ and $\delta_R = 0.7$ seem to clearly set the boundary for most states (with a few exceptions due to the errors). If a state is observed whose $V_D$ and $V_R$ values violate these thresholds, the state can be flagged as a dead-end with high probability. Setting a lower threshold can help to raise flags earlier on, when the conditions are of high-risk, but it is still not too late. We can apply the same thresholds to further flag high-risk actions (not shown). Lastly, we note (T1) from Theorem 1. It can be seen that only for all the yellow area (aside from the few erroneous states), $V_D = -1$. Clearly, no dead-end state can be a rescue, as seen by $V_R = 0$ for the yellow area too.

## 4   Empirical Setup for Dead-end Analysis

**Data:** We use DeD to identify medical dead-ends in a cohort of septic patients drawn from the MIMIC (Medical Information Mart for Intensive Care) - III dataset (v1.4) [22, 48]. This cohort totals 19,611 patients (17,730 survivors and 1,881 nonsurvivors), with 44 observation variables, and 25 treatment choices (5 discrete levels for each of IV fluid and vasopressor). We follow prior work [25] and aggregate each variable every four hours using the per-patient variable mean if data is present, or impute using the value from the nearest neighbor.

**Terminal States.** In our ICU setting, possible terminal states are either patient recovery (discharge from ICU) or death. We define "death" as the last recorded point in the EMR of nonsurviving patients when expiration is imminent, but may not necessarily be the biological point of death. In practice this definition of terminal state may occur hours or days before biological death and covers situations where care support devices are disconnected, when a patient requests a cessation of treatment, etc.

Our goal is to identify all *medical dead-end* states, defined as patient states from which death is unavoidable, regardless of future treatments. Relatedly, we also desire to discover all treatments that may possibly lead to a medical dead-end state in order to learn which treatments to avoid.

**SC-Network.** As observations of patient health are inherently partial, we need an informative latent representation of state [49], sufficient for evaluating treatment security. To form these state representations we process a sequence of observations prior to and including any time $t$ as well as the last selected treatment to form the state $s_t$. We train a standalone State Construction (SC) network using Approximate Information State (AIS) [50] in a self-supervised manner for this purpose. Details of AIS and how it is used to train the SC-Network are included in Appendix A4.

**D-Network and R-Network.** Computed states are given as input to the D- and R-networks to approximate $Q_D^*$ and $Q_R^*$. We use the double-DQN algorithm [51] to train each network (details included in Appendix A5). The outputs of trained D- and R- Networks produce value estimates of both the embedded patient state and all possible treatments to evaluate the probability of transitioning to a dead-end. This process of determining possibly high-risk treatments is central to DeD.

**Training:** We train the SC-, D-, and R- networks in an offline manner, using retrospective data (Fig A2). All models are trained with 75% of the patient cohort (14,179 survivors, 1,509 nonsurvivors),

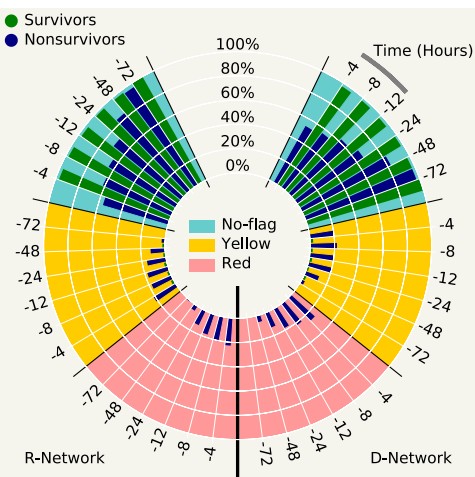

Figure 2: **Flag emergence for ICU patients.** Histograms of median $Q$ according to the flag status, for both surviving (green) and non-surviving patients (blue) according to the R-Network (left) and D-Network (right). The bars are plotted according to the time prior to the recorded terminal state (the maximum trajectory length is 72 hours) and measure the percentage of patients whose states raise either a red, yellow or no flag. There is a clear worsening trend of state values for nonsurviving patients as they approach their terminal state, beginning as early as 48 hours prior.

validated with 5% (890 survivors, 90 nonsurvivors), and we report all results on the remaining held out 20% (2,660 survivors, 282 nonsurvivors). Further details of how the patient cohort is processed are provided in Appendix A6. Finally, to mitigate the data imbalance between surviving and non-surviving patients we use an additional data buffer that contains only *the last transition* of nonsurvivors trajectories. Thus, a stratified minibatch of size 64 is constructed of 62 samples from the main data, augmented with 2 samples from this additional buffer, all selected uniformly. This same minibatch structure is used for training each of the three networks. For the training details see Appendix A4 and A5.

## 5 Empirical Results

### 5.1 Septic Dead-End State Prediction

**Experiment.** In order to flag potentially non-secure treatments, we examine if $Q_D$ and $Q_R$ of each treatment at a given state pass certain thresholds $\delta_D$ and $\delta_R$, respectively. To flag potential dead-end states, we need to probe the state values, for which we examine the *median* of $Q$ (rather than *max* of $Q$) against similar thresholds. Using the median helps to avoid extreme approximation error due to generalization from potentially insufficient data. We found that $\delta_D = -0.25$ and $\delta_R = 0.75$ minimize both false positives and false negatives, and use these as the thresholds for raising "red" flags. We also define a second, looser threshold of $\delta_D = -0.15$ and $\delta_R = 0.85$, as raising "yellow" flags with higher sensitivity but increased false positives. This looser threshold targets an early indication of a patient's health condition deteriorating toward a dead-end state. In Appendix Fig. A5 we report histogram of values at different quantiles, from which we established these thresholds.

**Results.** Using the specified thresholds, DeD identifies increasing percentages of patients raising fatal flags as nonsurvivors approach death (Figure 2 and Appendix Table A3). Note the distinctive difference between the trend of values in survivors (green bars) and nonsurvivors (blue bars). Over the course of 72 hours in the ICU, survivor trajectories raise nearly no red flag for both networks. In contrast, nonsurvivor trajectories demonstrate a steep reduction in *no-flag* zone with increasing numbers of patients flagged in the *Red* zone. The *Yellow* zone is dominated largely by the nonsurvivors, yet there are also survivors who ultimately recover. Under the red-flag threshold, more than 12% of treatments administered to non-surviving patients are identified to be detrimental 24 hours prior to death with a 0.6% false positive rate (Appendix Table A3). We further identify that 2.7% of non-surviving patient cases have entered unavoidable dead-end trajectories up to 48 hours before recorded expiration, with only a 0.2% false positive rate, i.e., patients misidentified as near death.

We find that 5% of nonsurviving patients maintain the red flag for their last 24 hours recorded in the ICU before reaching a death terminal state. This monotonically increases to 13.9% for patients who maintain a red flag through their final 8 hours of care (Appendix Fig. A6**b,c**). These patients likely reached a dead-end with no subsequent chance of recovery; this is as compared to 89.3% of nonsurviving test patients with no flag raised in their first 8 hours (Appendix Fig. A6**d**).

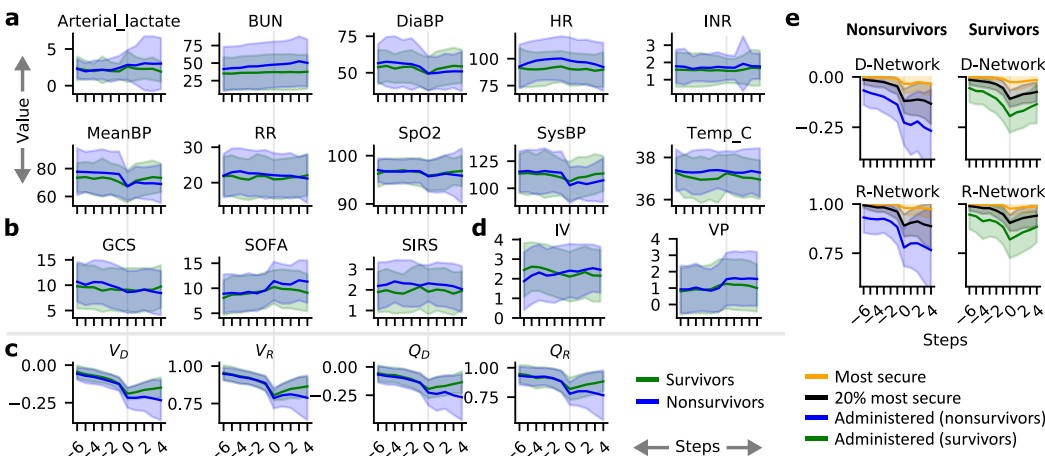

Figure 3: **Trend of measures around the first raised flag.** Various measures are shown 24 hours (6 steps) before the first (red or yellow) flag is raised and 16 hours (4 steps) afterward. All nonsurviving (blue) and surviving (green) patient trajectories that fall within this window are averaged, shaded areas represent a single standard deviation. (**a**) selected key vital measures and lab tests, (**b**) established clinical measures, and (**c**) DeD value measures of state ($V$) and administered treatment ($Q$) from the D- and R-Networks and, (**d**) administered treatments. There is a clear turning point 4 to 8 hours prior to the flag being raised, which precisely corresponds to a drastic increase of VP and IV treatments. (**e**) the value of the maximum, the 5th maximum (20% best) and the actually administered treatment, demonstrating that better treatments were available when the chosen treatments were administered.

There is a distinct difference between remaining on a flag for survivors and nonsurvivors (Appendix Fig. A6a). Even with our red threshold, very few survivors (0.5%) raise and remain on red-flag for more than eight hours, decreasing to nearly zero for longer periods. In contrast, 32.6% of nonsurvivors remain on red flags for similar duration with a fat tail. These results suggest that red-flag membership for long periods strongly correlates with mortality, inline with our theoretical analysis.

## 5.2 First Flag Analysis

**Experiment.** To further support our hypothesis that dead-end states exist among septic patients and may be preventable, we align patients according to the point in their care when a flag is "first raised". We select all trajectories in the test data with at least 24 hours (6 steps) prior to the first flag and at least 16 hours (4 steps) afterwards (77 surviving and 74 nonsurviving patients). This window excludes patients with flags that occur either too early or too late. This allows for an investigation of the average trend of patient observations, administered treatments as well as the measures used in DeD over a sufficiently large window (Figure 3).

**Results.** The $V$ and $Q$ values estimated by DeD have similar behavior in survivors and nonsurvivors prior to the first flag, but values diverge after the flag is raised. Notably, the time step pinpointed by DeD to raise a flag corresponds to a similar diverging trend among various clinical measures, including SOFA and patient vitals (Figure 3**a**,**b**). This distinct behavior is also seen if looser threshold values are used for $\delta_D$ and $\delta_R$ (Appendix Fig. A8). After the flag is raised there is slight improvement in all value estimates, perhaps in response to the change in treatment. However the values of nonsurviving patient trajectories quickly collapse while survivors continue to improve.

The results of this analysis suggest two main points. First, DeD identifies a clear *critical point* in the care timeline where nonsurviving patients experiencing an irreversible deterioration in health. Second, there is a significant gap (Figure 3**e**) between the value of administered treatments and the 20% most secure ones (5 out of 25). The critical point appears to arise when a patient's condition shifts towards improvement or otherwise enters a dead-end towards expiration. Perhaps most notable is that there is a clear inflection in the estimated values 4 to 8 hours prior to a flag being raised. Signaling this shift in the inferred patient response to treatment and the resulting flag may be used to provide an early indicator for clinicians (more conservative thresholds may be used to signal earlier). The trend of survivors shows that there is still hope to save the patient at this point. Note that *all* these patients

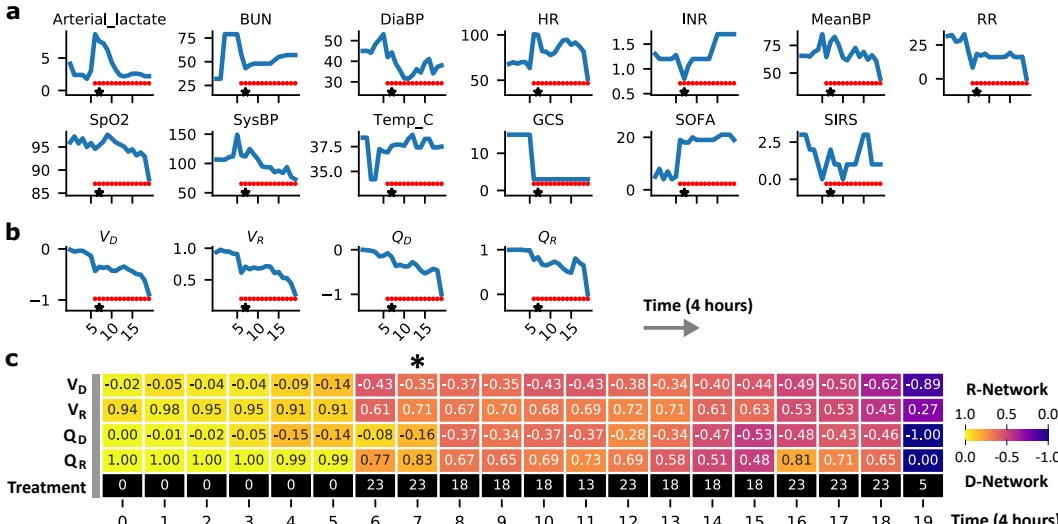

Figure 4: **Events at ICU.** Certain vitals (**a**) and DeD value measures from both D- and R-Networks (**b**) are shown for a non-surviving patient (ICU-Stay-ID 262011). The black asterisks demonstrate the presumed onset of sepsis at step 7, and the color dots corresponds to the raised red or yellow flags. Lastly, (**c**) illustrates steps along the patient's trajectory with DeD estimated values and selected treatments. Notably, from steps 5 to 6 the state has a sudden jump to a low-value region that it fails to escape from, aligned with significant inflections in the recorded vitals, approximately 4 to 8 hours before presumed onset of sepsis. (See Appendix Fig. A9 – Fig. A11 for full feature set plus accompanying excerpts from clinical notes of this and additional patients)

(survivors and nonsurvivors) are very similar in terms of both D/R values and their SOFA score prior to this point. This rejects the argument that survivors and nonsurvivors are inherently different. Additionally, while SOFA may appear correlated with DeD at the individual level, the trend of value functions can be noticeably more aggressive than SOFA with significantly less variance (Fig. A8). Further, most patients already have a high SOFA; hence, it is not sufficient for dead-end identification. DeD is however a provable methodology to this end. Figures 3**d** and **e** advocate that the choice of treatment may play a role in entering dead-ends, since the divergence/drop occurs before the flag. The gap in value between the administered treatments and those with the highest estimated security suggest that better treatments were available, even for patients who eventually recover (Figures 3**e**).

### 5.3 Individual Trajectories

**Experiment.** In our final analysis we extract relevant information surrounding a patient's value estimates from the electronic health record data, including the recorded clinical notes. We also use t-SNE [52] to project the state representations of the patient's trajectory, embedded using the SC-Network, among all recorded states in the test data (complete figures are presented in the Appendix).

**Results.** The clinicians' chart notes confirm existence of dead-ends with a noted need for intubation, hypotension, and a discussion of moving the patient's care to "comfort measures only" (Fig. A9**c**). Moreover, certain areas in the t-SNE projection of observed patient states appear to correspond with dead-end states (Fig. A9**b**). Notably, the dramatic shift of clinically established measures such as SOFA and GCS closely follow the decrease in DeD estimated values (Figure 4**a**, **b**). This is similar to the trends seen prior to raised flags (Figure 3). This qualitative analysis suggests that the estimates of $Q_D$ and $Q_R$ are reliable and informative, supporting our prior conclusions. Additional non-surviving patients are presented in Appendix Fig. A9 – Fig. A11.

## 6 Discussion

In this work, we have introduced an RL-based approach for learning what treatments *to avoid* based on observed patterns in limited offline data. We target avoiding treatments proportional to their chance

of leading to dead-ends, regions of the state space from which negative outcomes are inevitable. We establish theoretical results that expand the concept of dead-ends in RL, facilitating the notification of high-risk treatments or, as applied to healthcare, septic patient conditions with increased likelihood of leading to a dead-end. Globally, sepsis is a leading cause of mortality [27, 53, 54], and an important end-stage to many conditions. Consequently, even a slight decrease in mortality rate or improved efficacy of treatment could have a significant impact both in terms of saving lives and reducing costs.

Our work lays the groundwork for dead-end analysis in medical settings and is, to the best of our knowledge, the first use of RL to flag bad treatments rather than finding the best ones through estimating an optimal policy $\pi^*$. Our algorithm is generic, using RL methodology that is formally guaranteed to hold the security condition re-established in this paper. The discovery of dead-end states, and the treatments that likely lead to them, provides actionable insights in intensive care intervention. Further improvement of DeD's prediction quality could target additional features from the EMR environment, such as pre-ICU admission co-morbidities. In future work, we also hope to explore the specific drugs and dosages used in treatment.

Given its general construction, DeD is well matched for safety-critical applications of RL in data-constrained settings where it may be too expensive or unethical to collect additional exploratory data. With formal guarantees of satisfying the security condition, DeD is suitable for broader adoption when developing critical insights from retrospective data. Our framework is particularly relevant to data-constraint offline RL application domains such as robotics, industrial control, and automated dialogue generation where negative outcomes can be clearly identified [55].

**Limitations:** While DeD is a promising framework for decision support in safety-critical domains with limited offline data, there are certain core limitations. While we use median values of $Q_D$ or $Q_R$ to avoid extreme extrapolation, training the D- and R- networks is still performed offline and extrapolation is likely still occurring. For simplicity we did not estimate $Q_D$ or $Q_R$ with contemporary offline RL methods; however, DeD is generic and replacing the DDQN learning method with more recent approaches would be straightforward, which can significantly improve the pipeline (we also note that finding $Q_D$ or $Q_R$ is an *exponentially smaller* problem compared to finding $\pi^*$ to recommend best treatments). Additionally, we did not investigate the sensitivity of DeD to demographic information or with respect to specific features from the EMR. Thorough analysis of this sensitivity may elucidate the fairness and reliability of DeD. Finally, we did not externally validate DeD using data from a separate hospital or through investigation of suggested treatment avoidance by human clinicians. These investigations and more, concerning the causal entanglement of outcome and sequential treatments, are a focus of current and future work.

**Ethical Considerations and Societal Impact:** This work, or derivatives of it, should never be used in isolation to exclude patients from being treated, e.g., not admitting patients or blindly ignore treatments. The treatment-avoidance part of our proposed approach is meant to shrink the scope of possible treatment options, and help the doctors make better decisions. Signalling high-risk states is also meant to warn the clinicians for immediate attention before it possibly becomes too late. In both cases, the flags that DeD supplies are statistically tied to the training data and unavoidable sources of error and bias and should not be seen as a binary treat/don't treat decision. In particular, even in the case of red flags, the signals should not be interpreted as mathematical dead-ends with full precision. The intention of our approach is to assist clinicians by highlighting possibly unanticipated risks when making decisions and is not to be used as a stand-alone tool nor as a replacement of a human expert. Misuse of this algorithmic solution could carry significant risk to the well-being and survival of patients placed in a clinician's care.

The primary goal of this work is to establish a proof of concept where especially high-risk treatments can be avoided, where possible, in context of a patient's health condition. In acute care scenarios treatments come with inherent risk profiles and potential harms. In these settings tendencies to overtreat patients have arisen in attempt of ensuring their survival, increasing the chance of clinical errors to occur [56]. Recent clinical research has sought to simplify practice to only the most necessary treatments [4]. In this spirit, we seek to infer the long-term impact of each available treatment in view of their risk of pushing the patient into a medical dead-end. The secondary goal of our work, on the other hand, is signal when the patient's condition deteriorates, but may not be noticed by clinicians through monitoring clinical measures. This follows from the fact that DeD uses value functions, which provably enable such predictions.

---

[4]see http://jamanetwork.com/collection.aspx?categoryid=6017

## Acknowledgments and Disclosure of Funding

We thank our many colleagues who contributed to thoughtful discussions and provided timely advice to improve this work. We specifically appreciate the feedback provided by Nathan Ng, Sindhu Gowda, and the RL team at MSR Montreal, as well as the encouragement and suggested improvements provided by anonymous reviewers.

This research was supported in part by Microsoft Research, a CIFAR Azrieli Global Scholar Chair, a Canada Research Council Chair, and an NSERC Discovery Grant.

Resources used in performing this research were provided, in part, by Microsoft Research, the Province of Ontario, the Government of Canada through CIFAR, and companies sponsoring the Vector Institute www.vectorinstitute.ai/#partners.

## Data and Code Availability

Our code and pretrained models to replicate the analysis (including figures) presented in this paper is located at https://github.com/microsoft/med-deadend.

The MIMIC-III databases (DOI: 10.1038/sdata.2016.35) that support the findings of this study are publicly available through Physionet website: https://mimic.physionet.org, which facilitates reproducibility of the presented results. The cohort definition, extraction and preprocessing code can be found at https://github.com/microsoft/mimic_sepsis.

## Author Contributions

MF and MG designed the research. MF conceptualized theoretical ideas and developed formal results and proofs. JS developed the code for data generation and state construction and prepared initial experimental results. TK finalized the script to generate data from MIMIC, performed benchmarking of several state-construction algorithms–finalizing the decision to use AIS in the paper—and executed the experiments for the Life-Gate toy example. MF developed code for RL and made final experimental results. MF, TK and MG interpreted the results and wrote the manuscript. JS contributed to this work only during his internship at Microsoft Research. All the authors read and agreed on the final draft.

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
