## A1 Formal Results and Proofs

For simplicity, in all the arguments below, we refer to a positive terminal state as *recovery* and a negative terminal state as *death*. As with the main text, we also use *treatment* in place of *action*, which is the common term in RL texts. The rest of terminology follows the definitions presented in the main text.

**Lemma 1.**

L1.1. $V_D^*(s) = Q_D^*(s, a) = -1$ for all the treatments $a$ if and only if $s$ is a dead-end.

L1.2. $V_R^*(s) = \max_a Q_R^*(s, a) = 1$ if and only if $s$ is a rescue.

**Proof.** To prove the first part of the lemma, we assume $s$ is a dead-end and prove $Q_D^*(s, a) = -1$ for all the treatments. The definition of return directly implies that the return of all the trajectories from $s$ is precisely $-1$ since all of them reach a death terminal state and $\gamma = 1$. The expected return is therefore also $-1$ regardless of stochasticity; hence, $Q_D^*(s, a) = -1$.

Conversely, let for a given state $s$ we have $Q_D^*(s, a) = -1$ for all treatments $a$. We next prove that $s$ is a dead-end. For a transition $(s, a, s')$, the next state $s'$ is either of a non-terminal state with $r_D(s, a, s') = 0$, $max_{a'}Q_D^*(s', a') = -1$; a non-terminal state with $r_D(s, a, s') = 0$, $max_{a'}Q_D^*(s', a') > -1$; a death terminal state (i.e., $r_D(s, a, s') = -1$, $Q_D^*(s', a') = 0 \ \forall a'$); or a recovery terminal state (i.e., $r_D(s, a, s') = 0$, $Q_D^*(s', a') = 0 \ \forall a'$).

Let $C_R$ and $C_N$ denote respectively the sets of "recovery terminal states" and "non-terminal states $s'$ with $max_{a'}Q_D^*(s', a') > -1$". Note that $C_R$ and $C_N$ are disjoint, and that if a state $s'$ is not in $C_R \cup C_N$ then it is either a death terminal state (hence $r_D(\cdot, \cdot, s') = -1$ and $Q_D^*(s', \cdot) = 0$), or a non-terminal with -1 value (hence $r_D(\cdot, \cdot, s') = 0$ and $Q_D^*(s', \cdot) = -1$). Using Bellman equation, we write

$$-1 = Q_D^*(s, a) = \sum_{s'} T(s, a, s')[r_D(s, a, s') + \max_{a'} Q_D^*(s', a')]$$

$$= \sum_{s' \notin C_R \cup C_N} T(s, a, s') \times -1 + \sum_{s' \in C_R} T(s, a, s') \times 0 + \sum_{s' \in C_N} T(s, a, s') \max_{a'} Q_D^*(s', a')$$

$$= -\left[1 - \sum_{s' \in C_R \cup C_N} T(s, a, s')\right] + \sum_{s' \in C_N} T(s, a, s') \max_{a'} Q_D^*(s', a')$$

$$= -1 + \sum_{s' \in C_R} T(s, a, s') + \sum_{s' \in C_N} T(s, a, s') \left[1 + \max_{a'} Q_D^*(s', a')\right] \tag{4}$$

Because $T(s, a, s')$ is non-negative it therefore must be zero for both all $s' \in C_R$ and all $s' \in C_N$ (in the last line $\max_{a'} Q_D^*(s', a') \neq -1$). Hence, the next state is either a death terminal state or a non-terminal state with $Q_D^*(s', \cdot)$ of precisely $-1$ for all the treatments. Continuing with the same line of argument, it therefore follows that if $Q_D^*(s, a) = -1$ then all possible trajectories after $(s, a)$ will reach a death terminal state and all states on such trajectories assume the value of -1. Finally, if $V_D^*(s) = -1$ then $max_a Q_D^*(s, a) = -1$, which implies $Q_D^*(s, a) = -1$ for all $a$. It therefore follows that all trajectories from $s$ will reach a death terminal state, and by definition $s$ is a dead-end.

To prove L1.2, for the sufficiency we cannot use a similar argument as for L1.1 since not all the returns are $+1$; only the maximum needs to be $+1$. If $s$ is a rescue state, then by definition there must exist at least one trajectory w.p.1 to recovery. Starting from the last state before recovery on such a trajectory, we go backward and invoke Bellman equation. For the last state-treatment $(s'', a'')$ that transitions to recovery we have $Q_R^*(s'', a'') = +1$, hence $max_{a'}Q_R^*(s'', a') = +1$. Similarly, for all other states $s'$ on the deterministic trajectory to recovery, we conclude that $max_{a'}Q_R^*(s', a') = +1$, which implies $max_{a'}Q_R^*(s, a') = +1$, as stated in the lemma.

For the necessity, the argument is similar to that of L1.1. In particular, we can show in a similar way as in L1.1 that if $Q_R^*(s, a) = 1$ then $\max_{a'} Q_R^*(s', a') = 1$ for all the immediate next states $s'$ after $(s, a)$ if they are non-terminal. It implies that if $s'$ is non-terminal, then at least one treatment exists whose value at $s'$ is $+1$. Furthermore, for all state-treatment pairs whose values are $+1$, if the treatment causes transitioning to a terminal state it deterministically must be recovery (i.e., it cannot be recovery w.p. $p$ and death w.p. $1 - p$). We therefore conclude that there is at least one trajectory from $s$ to recovery with probability one; hence $s$ is a rescue state. ∎

**Lemma 2.** Let treatment $a$ be administered at state $s$, and $F_D(s, a)$ and $F_R(s, a)$ denote the probability that the next state will be terminal with death or recovery, respectively. Let further $P_D(s, a)$ and $P_R(s, a)$ denote the probability of transitioning to a dead-end or a rescue state, respectively, i.e. $P_D(s, a) = \sum_{s' \in \mathcal{S}_D} T(s, a, s')$ and $P_R(s, a) = \sum_{s' \in \mathcal{S}_R} T(s, a, s')$. Let $M_D(s, a)$ be the probability that the next state is neither a dead-end nor immediate death, and the patient ultimately expires while all the treatments are selected according to the greedy policy with respect to $Q_D^*$. Similarly, let $M_R(s, a)$ be the probability that the next state is neither immediate recovery nor a rescue state, but the patient ultimately recovers while all future treatments are selected according to the greedy policy with respect to $Q_R^*$. We have

L2.1. $-Q_D^*(s, a) = P_D(s, a) + M_D(s, a) + F_D(s, a)$

L2.2. $Q_R^*(s, a) = P_R(s, a) + M_R(s, a) + F_R(s, a)$

**Proof.** For the first part, Bellman equation reads as the following:

$$Q_D^*(s, a) = \sum_{s'} T(s, a, s')[r_D(s, a, s') + \max_{a'} Q_D^*(s', a')] \tag{5}$$

The next state $s'$ is either of the following:

1. a dead-end state, where $r_D(s, a, s') = 0$; $Q_D(s', a') = -1$, $\forall a'$ (due to Lemma 1); and $\sum_{s'} T(s, a, s') = P_D(s, a)$,
2. a death terminal state, where $r_D(s, a, s') = -1$; $Q_D(s', a') = 0$, $\forall a'$; and $\sum_{s'} T(s, a, s') = F_D(s, a)$,
3. a recovery terminal state where $r_D(s, a, s') = 0$, and $Q_D(s', a') = 0$, $\forall a'$, and
4. a non-terminal, non dead-end state, where $r_D(s, a, s') = 0$.

Item 3 vanishes and items 1 and 2 result in the first and the last terms in L2.1. For the item 4 above, assume any treatment $a'$ at the state $s'$ and consider all the possible roll-outs starting from $(s', a')$ under execution of the greedy policy w.r.t. $Q_D^*$ (which maximally avoids future mortality). At the end of each roll-out, the roll-out trajectory necessarily either reaches death with the $\mathcal{M}_D$ return of $-1$ for the trajectory, or it reaches recovery with the $\mathcal{M}_D$ return of $0$ for the trajectory. Hence, the expected return from $(s', a')$ will be $-1$ times the sum of probabilities of all the roll-outs that reach death (plus zero times sum of the rest). That is, $Q_D^*(s', a')$ is *the negative total probability of future death* from $(s', a')$ if optimal treatments (w.r.t. $Q_D^*$) are always known and administered afterwards. Consequently, $\max_{a'} Q_D^*(s', a')$ would be *negative minimum probability of future death* from state $s'$, again if optimal treatments are known and administered at $s'$ and afterwards. Therefore, $\sum_{s'} T(s, a, s') \max_{a'} Q_D^*(s', a')$ is negative minimum probability of future death from $(s, a)$ under optimal policy, which by definition is $-M_D(s, a)$. This shapes the middle term of L2.1 and concludes the proof.

The second part follows a similar argument. In particular, $Q_R^*(s', a')$ is the probability of reaching recovery under the execution of greedy policy w.r.t. $Q_R^*$ (which itself maximizes reaching a recovery terminal). Therefore, $\max_{a'} Q_R^*(s', a')$ is the maximum probability of reaching recovery under optimal policy from $s'$, and finally $\sum_{s'} T(s, a, s') \max_{a'} Q_R^*(s', a')$ induces maximum probability of reaching recovery from $(s, a)$.

■

**Lemma 3.**

L3.1. State $s$ is a dead-end if and only if $P_D(s, a) + F_D(s, a) = 1$ for *all* treatments $a$.

L3.2. State $s$ is a rescue if and only if $P_R(s, a) + F_R(s, a) = 1$ for *at least one treatment* $a$.

**Proof.** For part one, we note that $P_D(s, a)$, $M_D(s, a)$, and $F_D(s, a)$ are parts of the transition probability to the next state, hence

$$P_D(s, a) + M_D(s, a) + F_D(s, a) \leq 1$$

Therefore, $P_D(s, a) + F_D(s, a) = 1$ deduces $P_D(s, a) + M_D(s, a) + F_D(s, a) = 1$ (i.e., $M_D(s, a) = 0$). Invoking L2.1 induces $Q_D^*(s, a) = -1$ for all $a$; hence, $s$ is a dead-end due to L1.1. Conversely, if $s$ is a dead-end, L1.1 induces that $Q_D^*(s, a) = -1$ for *all treatments* $a$. Invoking (4) again, it follows that the next state cannot be a recovery terminal state or a non-terminal state with $\max_{a'} Q_D^*(s', a') > -1$, which implies the next state is either a dead-end or a death terminal state. Hence, $P_D(s, a) + F_D(s, a) = 1$ for all treatments $a$.

Similar proof holds for L3.2. Note that the counterpart of (4) for this case is as the following:

$$1 = 1 - \left( \sum_{s' \in C'_D} T(s, a, s') + \sum_{s' \in C'_N} T(s, a, s') \left[ 1 - \max_{a'} Q^*_R(s', a') \right] \right) \tag{6}$$

with $C'_D$ and $C'_N$ denoting, respectively, the sets of death terminal states and non-terminal states with $\max_{a'} Q^*_R(s', a') < 1$. Similarly, (6) necessitates $T(s, a, \cdot)$ must be zero for all transitions to $C'_D$ and $C'_N$.

■

**Theorem 1.** The followings hold:

T.1   $P_D(s, a) + F_D(s, a) = 1$ if and only if $Q^*_D(s, a) = -1$.

T.2   $P_R(s, a) + F_R(s, a) = 1$ if and only if $Q^*_R(s, a) = 1$.

T.3   There exists a threshold $\delta_D \in (-1, 0)$ independent of states and treatments, such that $Q^*_D(s, a) \geq \delta_D$ for all $s$ and $a$, unless if and only if $P_D(s, a) + F_D(s, a) = 1$.

T.4   There exists a threshold $\delta_R \in (0, 1)$ independent of states and treatments, such that $Q^*_R(s, a) \leq \delta_R$ for all $s$ and $a$, unless if and only if $P_R(s, a) + F_R(s, a) = 1$.

T.5   For any policy $\pi$, state $s$, and treatment $a$, if $\pi(s, a) \leq 1 + Q^*_D(s, a)$ and $\lambda \in [0, 1]$ exists such that $P_D(s, a) + F_D(s, a) \geq \lambda$, then $\pi(s, a) \leq 1 - \lambda$.

T.6   For any policy $\pi$, state $s$, and treatment $a$, if $\pi(s, a) \geq Q^*_R(s, a)$ and $\lambda \in [0, 1]$ exists such that $P_R(s, a) + F_R(s, a) \geq \lambda$, then $\pi(s, a) \geq \lambda$.

**Proof.** (T.1) and (T.2) are immediate from Lemma 1 and 3. For (T.3), it follows from (L1.1) that for a non-dead-end state $s$, we have $Q^*_D(s, a) > -1$. We choose $\Delta_D = \max_{s,a} [P_D(s, a) + M_D(s, a) + F_D(s, a)]$ for all non-dead-end and non-terminal states $s$ and all treatments $a$. If all the transition probabilities are stationary (or more generically, $\exists \lambda < 1 : T(s, a, s') < \lambda$ for all non-dead-end and non-terminal transitions) then $\Delta_D$ is a fixed value even though it might be very close to $-1$ in principle. As a result, it follows from L2.1 that for any threshold $\delta_D \in (-1, -\Delta_D]$ we have $Q^*_D(s, a) \geq -\Delta_D$ unless $s$ is a dead-end for which $Q^*_D(s, a) = -1$ due to L1.1. Furthermore, $\Delta_D$ only depends on the transition probabilities $T(s, a, s')$ and not the length of dead-ends. Similar proof concludes (T4).

In order to prove (T.5) and (T.6), we note that both $M_D(\cdot, \cdot)$ and $M_R(\cdot, \cdot)$ are non-negative for all state-treatments. Using the antecedent of (T.5), $P_D(s, a) + F_D(s, a) \geq \lambda$, as well as invoking Lemma 2, it yields:

$$\begin{aligned} Q^*_D(s, a) &\leq Q^*_D(s, a) + M_D(s, a) \\ &= -(P_D(s, a) + F_D(s, a)) \leq -\lambda \end{aligned}$$

which implies $1 + Q^*_D(s, a) \leq 1 - \lambda$. Hence, setting $\pi(s, a) \leq 1 + Q^*_D(s, a)$ deduces $\pi(s, a) \leq 1 - \lambda$.

Similarly, for (T.8) we have $P_R(s, a) + F_R(s, a) \geq \lambda$, therefore

$$\begin{aligned} Q^*_R(s, a) &\geq Q^*_R(s, a) - M_R(s, a) \\ &= P_R(s, a) + F_R(s, a) \geq \lambda \end{aligned}$$

As a result, $\pi(s, a) \geq Q^*_R(s, a)$ deduces $\pi(s, a) \geq \lambda$.

■

**Proposition 1**. Let $Q_D(s,a)$ be an approximation of $Q_D^*(s,a)$, such that

1. $Q_D(s,a) = Q_D^*(s,a) = -1$ for all $s \in \mathcal{S}_D$.
2. For all other states, the values satisfy monotonicity with respect to the Bellman operator $\mathcal{T}^*$, i.e. $Q_D(s,a) \leq (\mathcal{T}^*Q_D)(s,a)$ for all $(s,a)$.
3. All values of $Q_D(s,a)$ remain non-positive.

The security condition still holds if $\pi(s,a) \leq 1 + Q_D(s,a)$.

**Proof**. Using assumptions 1 and 2 we write

$$Q_D(s,a) \leq (\mathcal{T}^*Q_D)(s,a) \tag{7}$$
$$= \sum_{s'} T(s,a,s') \left[ r_D(s,a,s') + \max_{a'} Q_D(s',a') \right]$$
$$= - \sum_{s' \in \mathcal{S}_D} T(s,a,s') - \sum_{s' \in C'_D} T(s,a,s') + \sum_{s' \notin \mathcal{S}_D \cup C'_D} T(s,a,s') \left[ r_D(s,a,s') + \max_{a'} Q_D(s',a') \right] \tag{8}$$
$$= -P_D(s,a) - F_D(s,a) - \beta_D(s,a) \tag{9}$$

in which, $-\beta_D$ is the last term of (8). The reward of $\mathcal{M}_D$ is always zero unless at death terminal states where $r_D(s,a,s') = -1$. Hence, assumption 3 implies that $\beta_D(s,a)$ is always non-negative, regardless of how much $Q_D(s',a')$ is inaccurate. The rest of argument in Theorem 1 remains valid with $Q_D$ and $\beta_D$ replacing $Q_D^*$ and $M_D$.

∎

**Remark 1.** One setting that holds assumption 3 of Proposition 1 is in the tabular case where each $Q_D(s,a)$ is stored separately and under the assumption that all $(s,a)$ pairs are initialized with any non-positive number (naturally in $[-1,0]$). In the general case involving non-tabular estimators, a practical way to assure that Assumption 3 of Proposition 1 holds is to clip all the values at $-1$ and $0$.

**Remark 2.** There are certain cases that formally satisfy assumption 2. For example, the true value of *any* policy (not necessarily optimal) satisfies this inequality [11]. Another example is in the tabular setting when all values are initialized *pessimistically* (e.g., at $-1$); however, pessimistic initialization may increase false positives because all unseen $(s,a)$ pairs will be inferred as dead-ends. In other cases, since $Q_D(s,a)$ is the convergence point of Bellman error, it is likely that for many state-treatment pairs assumption 2 holds. Nevertheless, one should note that this assumption needs further scrutiny and may not hold in general when function approximation is used. In particular, over-estimation issue (if exists for any state-treatment pair) will forfeit assumption 2.

**Remark 3.** Proposition 1 implies that under certain assumptions, at each state only the value of treatments that lead to dead-end states w.p.1 has to be fully converged. Importantly, such values are independent of the values of other (non-dead-end) states, since according to Lemma 3 a dead-end's next state is also always either a dead-end or a death terminal state, regardless of the administered treatment. In an abstract way, it leaves out the necessity of learning the value for all the resulting trajectories from other treatments at the initial state as well as in the future resulting trajectories, which grow exponentially. Hence, at least in the tabular case with $-1$ value-initialization, learning the treatment avoidance method by securing the behavioral policy is an exponentially smaller problem than learning optimal policy (or optimal values), which advises for best treatments.

**Remark 4.** Full convergence of values of dead-end states $\mathcal{S}_D$ to -1 in Assumption 1 can be relaxed to $-(1-\epsilon)$ for some $\epsilon \in [0,1)$. In that case, rewriting (9) induces that the security guarantee will degrade to $\pi(s,a) \leq 1 - (1-\epsilon)\lambda$. That is, for a risky treatment, abiding by $1 + Q_D$ guarantees less decrease of its probability than what the security conditions requires. This may be addressed by adjusting the thresholds more conservatively.

## A2 Further Remarks on Related Work

In light of discussions with reviewers during the rebuttal period, we feel the need to honor similarities and differences between our work and those introduced in Irpan et al. [46] more thoroughly than space constraints allow in the main body of this paper. While there are partial parallels in terms of grounding ideas, our theoretical development vastly diverges from Irpan et al. [46], which relies wholly on empirical exploration and is centered wholly on policy evaluation rather than the assessment of specific decisions an agent may make. We summarize key important differences as follows:

- Their concept of *feasible* is simply being non-catastrophic and is different from *rescue*, which is a state where recovery is reachable w.p.1. (i.e., there is no parallel for rescue states in their work).

- The properties of the Q function and how it formally links to the probabilities of a state being feasible or catastrophic is not derived, discussed, or used in their work.

- Their OPC metric is a proxy for evaluation/ranking learned policies. They do not use the framing to identify problematic or high-risk actions that may lead to catastrophic behavior. More accurately, there is no particular parallel for the concept of (treatment) security, its definition, and the formal guarantees which then shape the foundation of DeD.

- In their work, the classification component is used to identify the value of state-action pairs on a binary $\{0, 1\}$ scale. This makes negative behavior somewhat unidentifiable (they acknowledge this) from intermediate feasible states that do not correspond to terminal conditions.

- Our dead-end construction (reward of -1 for bad outcomes + no-discounting) provides an inherently different value function, which (with a negative sign) formally gives rise to the minimum probability of bad outcomes in the future.

- Side note: in dangerous and stochastic environments and for sufficiently long episodes, their Theorem 1 results in the trivial bound (since the lower-bound becomes a negative value). Their experiments are restricted to robotic tasks and the Atari game of Pong; thus, this core problem has remained hidden in their work.

At a high level both Irpan et al. [46] and our work exploit constructed asymmetries within the state space to identify regions that are undesirable and should be avoided. The notions of *feasible* and *catastrophic* in Irpan et al. [46] are related, in context of an optimal policy $\pi^*$, with $P_{\pi^*}(\text{success}|\text{feasible}) > 0$ where $P_{\pi^*}(\text{success}|\text{catastrophic}) = 0$ always. Thus, by being able to classify which states are *catastrophic* evaluation of any trajectory containing such states is made significantly easier when evaluating policies developed from observational data. Irpan, et al. worked to label all state-action pairs as either *feasible* or *catastrophic* using positively-unlabeled classification.

With a similar asymmetry, but generalized to encompass the delicate dynamics often observed in safety-critical domains, we formalize the relationship between the special states (described in Section 3.2) and the terminal conditions of death or recovery as follows: $P(\text{recovery}|\text{rescue}) = 1$ for some policy $\pi$ (including the optimal policy $\pi^*$). In contrast, dead-end states have a more extreme condition where $P(\text{death}|\text{dead-end}) = 1$ for all policies $\pi$. This helps to emphasize the importance of identifying treatments that may lead to dead-end states and subsequently influence decision-makers to avoid selecting those treatments. The means by which we infer the risk of a treatment (or action) is through a pair of independent MDPs used to identify the value of a state-treatment pair in accordance to its risk of being a dead-end or the chance it may lead to rescue and being a rescue state. This joint inference problem is used to affix and confirm whether a state should be avoided (and all treatments leading to this state) as discussed in Theorem 1 in Section 3.4.

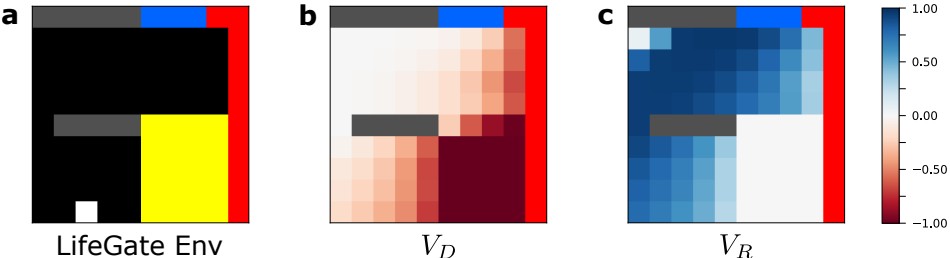

Fig. A1: **The Life-Gate Example.** The tabular navigation task of life-gate is illustrated in (**a**). Its corresponding dead-end and rescue (nearly optimal) state-value functions, $V_D^*$ and $V_R^*$, ares shown in (**b**) and (**c**), respectively. The value functions are learned through Q-learning and with the definition of $\mathcal{M}_D$ and $\mathcal{M}_R$.

## A3 Toy Problem: LifeGate

In this section we present a detailed toy-example with tabular state-space, called Life-Gate (Fig. A1). The white square depicts the agent's position, which has five actions corresponding to moving up, down, left, right, and doing nothing (no-up). The grey walls are neutral barriers, hitting to which does not have any effect, yet the agent cannot pass them. There are two possible terminal states: (1) death gates (shown in red) and (2) life gates (shown in blue), and the goal is for the agent to reach a life gate. If the agent lies in black areas, any action will cause a forceful move to the right with DEATH-DRIFT $= 40\%$ probability, or otherwise performs a cardinal move as expected. On the other hand, the yellow areas are all *dead-end* states. If the agent reaches any of the yellow positions, at each step afterwards, the agent will move to the right with $70\%$ probability or remain put with $30\%$ probability, regardless of the taken action. Hence, the agent will be on an inescapable dead-end trajectory to a death gate with random length. However, the agent will not see any of the colors and the state only comprises agent's x-y position.

We use Q-learning to compute the value functions of $\mathcal{M}_D$ and $\mathcal{M}_R$ as detailed in the main text: using discount of $\gamma = 1$ for both, and $\mathcal{M}_R$ only assigns the reward of $+1$ in the case of reaching a life gate (zero otherwise), while $\mathcal{M}_D$ assign the reward of $-1$ if transitioning to a death gate (and zero otherwise). We stop training before full convergence; hence, there are possible learning errors (e.g., upper left corner for $V_R^*$). Of note, the value of walls (which are all zero) is simply an artifact of choosing zero for initialization of the Q-tables.

This set-up comprises an interesting case. The agent faces an environment to explore, with no knowledge of possible dangers. Importantly, once a dead-end state is reached, it may take some random number of steps before reaching a death gate, where the agent would realise expiration. All along such trajectories of dead-end states, the agent still has to choose actions with the (false) hope of reaching a life-gate. Discovering any single dead-end state and signaling the agent when the state gets in the scope would be of significant importance. On the other hand, the adjacent states to dead-ends are possibly the most critical ones to alert, as it might be the last chance to still do something.

Let us probe this problem with the tools provided by Theorem 1. Based on Theorem 1, there are thresholds $\delta_D$ and $\delta_R$ which completely separate the values of dead-end states from the rest, both in term of $V_D^*$ and $V_R^*$. In this example, even with the errors due to lack of full convergence, $\delta_D = -0.7$ and $\delta_R = 0.7$ seem to clearly set the boundary for most states. There are some exceptions though. For example the top right corner is a false positive for $V_R^*$ due to learning errors, or at the top row of the yellow area, all the states are dead-ends but not all of them passes this test, again due to learning errors. If a state is observed whose $V_D$ and $V_R$ values violate these thresholds, the state can be flagged as a dead-end with high probability. Setting a lower threshold can help to raise flags earlier on, when the conditions are becoming high-risk, but it is not too late. We can see that $\delta_D = -0.2$ can act as a early-warning flag. Lastly, to also see (T1) and (T2) of Theorem 1, we note that only for all the yellow area (setting aside the few erroneous states), $V_D = -1$ and $V_R = 0$.

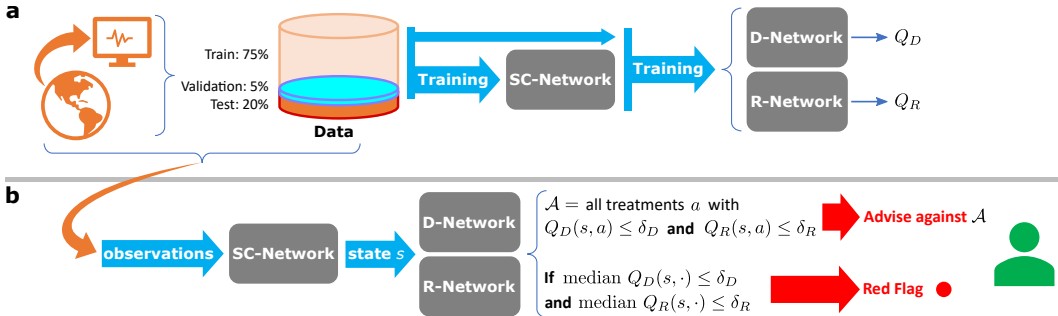

Fig. A2: **Dead-ends Discovery (DeD)**. Our pipeline includes two phases: (**a**) Training phase: using real-world data, we train the three neural networks to set-up i) state construction (SC-Network), ii) dead-end values (D-Network) and iii) rescue values (R-Network). (**b**) Test phase: the trained networks are used to map the immediate history of observations and the last action into $Q_D$ and $Q_R$ to infer risky conditions and dead-end outcomes, which is passed to the human decision-maker.

## A4    State Construction Details and Training

This section highlights the construction and development of the state construction network used to embed the observation sequences of a patient's health condition into a state representation to be used in the reinforcement learning networks used for the detection and avoidance of dead-end states.

### A4.1    Notation

Let $\mathcal{D} = \{\tau_j\}_{j=1}^n$ denote the batch data of $n$ trajectories obtained from the database of patients with sepsis in the intensive care usit. We assume that this data is generated from a time-homogeneous partially observable Markov decision process (POMDP). Each trajectory $\tau_j$ has a finite number of transitions $m_j$. Each transition in a trajectory $j$ is a tuple with four entries $(O_{t,j}, A_{t,j}, R_{t,j}, O_{t+1,j})$, where $j \in \{1, \ldots, n\}$, $t \in \{1, \ldots, m_j\}$. The observation and action (treatment) spaces are defined as in Sec. A6 where:

- $O_{t,j}, O_{t+1,j} \in \mathcal{O}$ are the observations received at times $t$ and $t+1$ respectively in trajectory $j$ and $\mathcal{O} \subset \mathbb{R}^{d_{\mathcal{O}}}$ is the observation space. In our case for the sepsis treatment problem, the observation space is 44 dimensional.

- $A_{t,j} \in \mathcal{A}$ is the action taken at time $t$ in trajectory $j$ and $\mathcal{A}$ is the action space. In this work we restrict attention to discrete action spaces of finite cardinality, $|\mathcal{A}| = n_a$. In our case for the sepsis treatment problem, $n_a = 25$.

- $R_{t,j} \in \mathbb{R}$ is the per-step reward received at time $k$ in trajectory $j$. We use an end-of-trajectory binary reward signal of $\pm 1$ (we, however, do not explicitly make use of the reward in the state construction network because we only focus on state representation learning for dynamics prediction).

For clarity we drop the trajectory index $j$ throughout the remainder of this section unless it is necessary to differentiate between trajectories. Let $\hat{d}_{\mathcal{S}}$ denote the dimension of the learned state representation ($\hat{S}$), which is a hyper-parameter that needs to be chosen. Our objective is to learn a state construction function $\psi : \{O_{0:t}, A_{0:t-1}\} \mapsto \hat{S}_t$, $t \geq 1$, and $\hat{S}_t \in \hat{\mathcal{S}} \subset \mathbb{R}^{\hat{d}_{\mathcal{S}}}$. In addition to $\psi$, the approaches outlined in the next section also involve another function: a dynamics predictor $\phi$ that involves predicting the next observation $\hat{O}_{t+1}$. Hence, the function $\phi : \hat{\mathcal{S}} \times \mathcal{A} \to \Delta(\mathcal{O})$, where $\Delta(x)$ denotes a probability distribution of $x$, estimates the conditional distribution of the next observation given the current state representation and action.

### A4.2    State Construction (SC) Network

We construct the state representation of a patient's condition by training a set of coupled functions, as motivated by the Approximate Information State (AIS) approach [50]. AIS satisfies two key properties: 1) each state is "Markovian" or sufficient for the prediction of the next state, and 2) observations are distinguishable when mapped

to their corresponding states if they result in different future trajectories. The first function, denoted by $\psi$, encodes the observed sequence patient conditions and the treatments administered into a compressed representation. This representation (corresponding to the state used in the reinforcement learning networks) is then passed, along with the current treatment, to a decoding function $\phi$ to predict the next patient observation.

The input to $\psi$ is the concatenation of the observation $O_t$ and last selected action $A_{t-1}$. For the function $\psi$ we use a 3-layer Recurrent Neural Network (RNN), where the first layer is a fully connected layer that maps the current observation and action (69 dimensional input: 44 dimensional observation with a 25 dimensional one-hot encoded action) to 64 units with ReLU activation. This is followed by another $(64, 128)$ fully connected layer with ReLU activation which is followed by a gated recurrent unit [57] layer with hidden state size $\hat{d}_\mathcal{S}$. The output of this recurrent layer is used as the state representation $\hat{S}_t$. The current action $A_t$ is concatenated to the state representation $\hat{S}_t$ and then fed through the decoder function $\phi$ to predict the next observation $\hat{O}_{t+1}$. The function $\phi$ is comprised of a three layer neural network with sizes $(\hat{d}_\mathcal{S} + 25, 64)$, $(64, 128)$ and $(128, 44)$ (with ReLU activation for the first two layers). The last layer outputs a 44-dimensional vector, which forms the mean vector of a unit-variance multivariate Gaussian distribution, samples from which are used to predict the next observation. A schematic of the the state construction network is provided in Fig. A3. The two functions $\psi$ and $\phi$ that comprise the state construction network are jointly trained by maximizing the negative log likelihood of the predicted next observation $\hat{O}_{t+1}$.

This is formulated by maximizing the objective:

$$\mathcal{L}(\mathcal{O}_{t+1}, \hat{\mathcal{O}}_{t+1}) = -\sum^{d_{\mathcal{O}_j}} \log \mathcal{N}(\mathcal{O}_{t+1,j}; \mu_j, \sigma_j^2)$$

where $\mu_j = \hat{O}_{t+1}, \sigma_j^2 = 1$, and $\hat{\mathcal{O}}_{t+1} = \psi(\phi(O_t, A_{t-1}), A_t)$.

### A4.3 Hyperparameter selection

The dimension of the state representation $\hat{d}_\mathcal{S}$ was chosen from among $\{4, 8, 16, 32, 64, 128, 256\}$ dimensions. The choices of the size of neural network layers was chosen proportional to the size of $\hat{d}_\mathcal{S}$, with the final values reported in the prior subsection following the optimal choice of $\hat{d}_\mathcal{S}$ being equal to 64. The model construction network was trained for 600 epochs with learning rates of $\{0.0001, 0.0005, 0.001, 0.005, 0.001\}$ with the choice of $lr = 0.0005$ providing the optimal training of the network. We demonstrate the evaluation of the choice of the dimension for the state representation in Fig. A4.

## A5   D- and R-Networks Training Details

We use double-DQN algorithm to train both networks. We refer the reader to our code for the implementation details (and we tried to make the code straightforward and relatively easy to understand). In particular, both D- and R-Networks consist of two linear layers with 64 nodes. The first layer is followed by ReLU nonlinearity and the second layer directly outputs 25 nodes corresponding to the 25 treatments. We use learning rate of $0.0001$ and minibatch size of $64$. In each minibatch, we select 62 transitions uniformly from the train data and append it with two uniformly selected "death" transitions (last transitions of nonsurvivor patients). All other chosen hyper-parameters can be found in the *config.yaml* file in the root directory of our code.

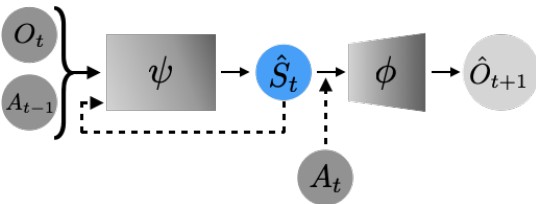

Fig. A3: The state construction network, comprised of the encoding function $\psi$ that provides the state representation $\hat{S}_t$ that is used with the decoding function $\phi$ to predict the next observation $\hat{O}_{t+1}$.

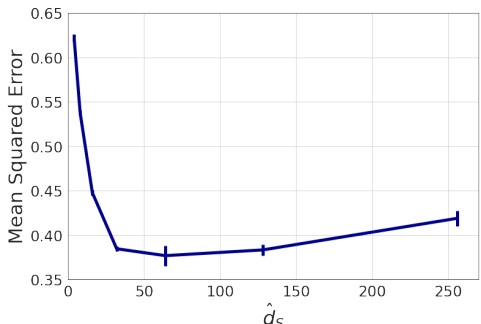

Fig. A4: Analysis of setting the dimension of the learned state representation $\hat{d}_\mathcal{S}$ and its effect on the accuracy of predicting the next observation. The bars represent standard deviation. With this, we determine to set $\hat{d}_\mathcal{S} = 64$ in the SC-network.

## A6    Data Details

We use the MIMIC (Medical Information Mart for Intensive Care) - III dataset (v1.4), which has been sourced from the Beth Israel Deaconess Medical Center in Boston, Massachusetts [22, 48]. This dataset comprises of deidentified patient treatment records of patients admitted to critical care units (CCU, CSRU, MICU, SICU, TSICU). The database includes data collected from 53,423 distinct hospital admissions of patients over 16 years of age for a period of 12 years from 2001 to 2012. The MIMIC dataset has been used in many reinforcement learning for health care projects, including mechanical ventilation and sepsis treatment problems. There are various preprocessing steps that are performed on the MIMIC-III dataset in order to obtain the cohort of patients and their relevant observables for the sepsis treatment study.

To extract and process the data, we follow the approach described in [25] and the associated code repository given in [58]. This includes all ICU patients over 18 years of age who have some presumed onset of sepsis (following the Sepsis 3 criterion) during their initial encounter in the ICU after admission, with a duration of at least 12 hours. These criteria provide a cohort of 19,611 patients, among which there is an observed mortality rate just above 9%, where mortality is determined by patient expiration within 48h of the final observation. Observations are processed and aggregated into 4h windows with treatment decisions (administering fluids, vasopressors, or both) discretized into 5 volumetric categories. All data is normalized to zero-mean and unit variance and missing values are imputed using k-Nearest Neighbor imputation, where possible. In the absence of similar observations any remaining missing values filled with the population mean. We report the 44 features used for the Dead-end approach proposed in this paper in Table A1 with high-level statistics for the extracted cohort in Table A2.

Table A1: Patient features used for learning state representations for predicting future observations

| Age | Gender | Weight (kg) | Re-admission |
|---|---|---|---|
| Glasgow Coma Scale | Heart Rate | Sys. BP | Dia. BP |
| Mean BP | Respiratory Rate | Body Temp (C) | FiO2 |
| Potassium | Sodium | Chloride | Glucose |
| INR | Magnesium | Calcium | Hemoglobin |
| White Blood Cells | Platelets | PTT | PT |
| Arterial pH | Lactate | PaO2 | PaCO2 |
| PaO2 / FiO2 | Bicarbonate (HCO3) | SpO2 | BUN |
| Creatinine | SGOT | SGPT | Total Bilirubin |
| Output (4h) | Output (total) | Cumulated Balance | SOFA |
| SIRS | Shock Index | Base Excess | Mech. Ventilation |

Table A2: MIMIC Sepsis Cohort Statistics

| Variable | MIMIC ($n = 19611$) | Variable | MIMIC ($n = 19611$) |
|---|---|---|---|
| **Demographics** | | **Outcomes** | |
| Age, years | 66.2 (53.8-78.1) | Deceased | 1881 (9.6%) |
| Age range, years | | Vasopressors administered | 5664 (28.9%) |
| 18-29 | 741 (3.8%) | Fluids administered | 17812 (90.8%) |
| 30-39 | 896 (4.6%) | Ventilator used | 9353 (47.7%) |
| 40-49 | 2029 (10.3%) | | |
| 50-59 | 3471 (17.7%) | **Severity Scores** | |
| 60-69 | 4321 (22.0%) | SOFA | 5 (3.0-8.0) |
| 70-79 | 4086 (20.8%) | SIRS | 2 (1.0-2.0) |
| 80-89 | 3069 (15.6%) | Shock Index | 0.72 (0.6-0.86) |
| $\geq$90 | 998 (5.1%) | | |
| Gender | | | |
| Male | 10917 (55.6%) | | |
| Female | 8694 (44.3%) | | |
| Re-admissions | 1424 (7.3%) | | |
| | | | |
| **Physical exam findings** | | | |
| Temperature ($^\circ$C) | 37.2 (36.6-37.7) | | |
| Weight (kg) | 79.7 (66.7-95.2) | | |
| Heart rate (beats per minute) | 86.0 (75.0-98.0) | | |
| Respiratory rate (breaths per minute) | 19.8 (16.6-23.3) | | |
| Systolic blood pressure (mmHg) | 118.3 (105.8-133.6) | | |
| Diastolic blood pressure (mmHg) | 56.6 (48.6-65.4) | | |
| Mean arterial pressure (mmHg) | 77.0 (69.0-86.7) | | |
| Fraction of inspired oxygen (%) | 40.0 (35.0-50.0) | | |
| P/F ratio | 307.5 (192.0-579.0) | | |
| Glasgow Coma Scale | 14.8 (11.0-15.0) | | |
| | | | |
| **Laboratory findings** | | | |
| Hemotology | | Coagulation | |
| White blood cells (thousands/$\mu$L) | 10.8 (7.7-14.8) | Prothrombin time (sec) | 14.3 (13.1-16.4) |
| Platelets (thousands/$\mu$L) | 202.0 (137.0-286.0) | Partial thromboplastin time (sec) | 32.6 (27.6-44.9) |
| Hemoglobin (mg/dL) | 10.2 (9.1-11.4) | INR | 1.3 (1.1-1.5) |
| Base Excess (mmol/L) | 0.5 (0.0-2.6) | | |
| Chemistry | | Blood gas | |
| Sodium (mmol/L) | 138.9 (136.0-141.0) | pH | 7.41 (7.35-7.44) |
| Potassium (mmol/L) | 4.0 (3.7-4.4) | Oxygen saturation (%) | 97.3 (95.5-98.8) |
| Chloride (mmol/L) | 105.0 (101.0-108.5) | Partial pressure of O2 (mmHg) | 124.0 (85.0-241.1) |
| Bicarbonate (mmol/L) | 25.0 (22.0-28.0) | Partial pressure of CO2 (mmHg) | 40.6 (36.0-46.0) |
| Calcium (mg/L) | 8.3 (7.8-8.8) | | |
| Magnesium (mg/L) | 2.0 (1.8-2.2) | | |
| Blood urea nitrogen (mg/dL) | 22.0 (14.0-36.0) | | |
| Creatinine (mg/dL) | 1.0 (0.7-1.5) | | |
| Glucose (mg/dL) | 127.4 (107.0-156.0) | | |
| SGOT (units/L) | 38.0 (22.0-74.0) | | |
| SGPT (units/L) | 30.0 (17.0-64.0) | | |
| Lactate (mg/L) | 1.5 (1.1-2.2) | | |
| Total bilirubin (mg/L) | 0.7 (0.4-1.5) | | |

# A7 Supporting Figures and Tables

**a Red flag thresholds**

|        | D-Network | | | | R-Network | | | | Full | | | |
|--------|-----------|---|-----------|---|-----------|---|-----------|---|---------|---|--------------|---|
|        | Survivors | | Nonsurvivors | | Survivors | | Nonsurvivors | | Survivors | | Nonsurvivors | |
|        | $Q_D$ | $V_D$ | $Q_D$ | $V_D$ | $Q_R$ | $V_R$ | $Q_R$ | $V_R$ | $Q$ | $V$ | $Q$ | $V$ |
| **-72 h** | 0.2% | 0.2% | 0.0% | 0.0% | 0.5% | 0.2% | 2.8% | 0.9% | 0.0% | 0.2% | 0.0% | 0.0% |
| **-48 h** | 1.2% | 0.4% | 8.1% | 5.4% | 1.5% | 0.5% | 5.9% | 4.3% | 0.7% | 0.2% | 2.7% | 2.7% |
| **-24 h** | 1.1% | 0.4% | 16.3% | 13.0% | 1.2% | 0.3% | 16.7% | 13.0% | 0.6% | 0.1% | 12.2% | 10.6% |
| **-12 h** | 0.9% | 0.4% | 20.2% | 18.2% | 0.7% | 0.3% | 20.2% | 17.4% | 0.4% | 0.2% | 12.8% | 14.7% |
| **-8 h** | 1.0% | 0.4% | 24.5% | 21.9% | 0.9% | 0.3% | 19.3% | 20.4% | 0.6% | 0.2% | 13.4% | 17.8% |
| **-4 h** | 1.2% | 0.5% | 29.7% | 26.4% | 0.7% | 0.5% | 24.9% | 22.7% | 0.5% | 0.3% | 20.1% | 22.0% |

**a Yellow flag thresholds**

|        | D-Network | | | | R-Network | | | | Full | | | |
|--------|-----------|---|-----------|---|-----------|---|-----------|---|---------|---|--------------|---|
|        | Survivors | | Nonsurvivors | | Survivors | | Nonsurvivors | | Survivors | | Nonsurvivors | |
|        | $Q_D$ | $V_D$ | $Q_D$ | $V_D$ | $Q_R$ | $V_R$ | $Q_R$ | $V_R$ | $Q$ | $V$ | $Q$ | $V$ |
| **-72 h** | 1.6% | 0.5% | 4.6% | 2.8% | 1.8% | 0.2% | 5.6% | 2.8% | 0.5% | 0.0% | 2.8% | 2.8% |
| **-48 h** | 3.1% | 2.2% | 12.4% | 11.9% | 2.7% | 2.1% | 14.1% | 11.9% | 1.6% | 1.5% | 10.3% | 9.7% |
| **-24 h** | 2.7% | 1.8% | 17.1% | 20.3% | 2.2% | 1.8% | 13.8% | 15.9% | 1.4% | 1.4% | 10.6% | 13.4% |
| **-12 h** | 3.3% | 2.5% | 19.4% | 19.4% | 3.4% | 2.4% | 17.4% | 17.8% | 2.0% | 1.7% | 15.1% | 17.1% |
| **-8 h** | 3.0% | 2.1% | 20.8% | 21.9% | 2.6% | 2.1% | 21.6% | 17.8% | 1.2% | 1.5% | 18.6% | 15.6% |
| **-4 h** | 3.2% | 2.4% | 20.1% | 20.1% | 3.0% | 2.4% | 16.8% | 21.2% | 1.7% | 1.5% | 16.1% | 17.6% |

Table A3: **Prediction of potentially life-threatening treatments and states (full list).** Similarly to 2, the results correspond to the part of test data that satisfies having minimum length of the corresponding time step (X hours before terminal). To raise a flag, a patient must concurrently violate the corresponding thresholds, as specified in 2. $Q$ columns correspond to the value of actually selected treatments, while $V$ columns correspond to the median value of patients' state at the corresponding time.

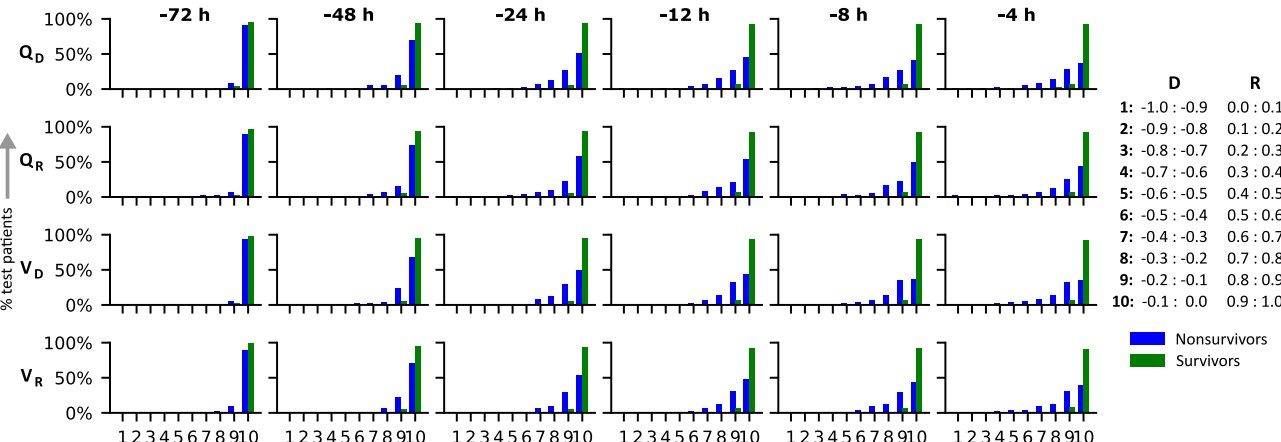

Fig. A5: **Full histogram of values in different time steps.** The histograms are plotted from the part of test data that satisfies having minimum length of the time step. The four rows are corresponding to the following: $V_D$ and $V_R$: median value of states from D-Network and R-Network, respectively, and $Q_D$ and $Q_R$: value of the selected treatments at the given time step from D-Network and R-Network, respectively. Note the distinctive difference between the trend of values in survivor (green bars) and nonsurvivor (navy bars) trajectories. In particular, in the course of 72 hours in the ICU, there is not much change in the value of selected or median treatment for the survivor patients, which is completely in contrast with those of nonsurvivor patients.

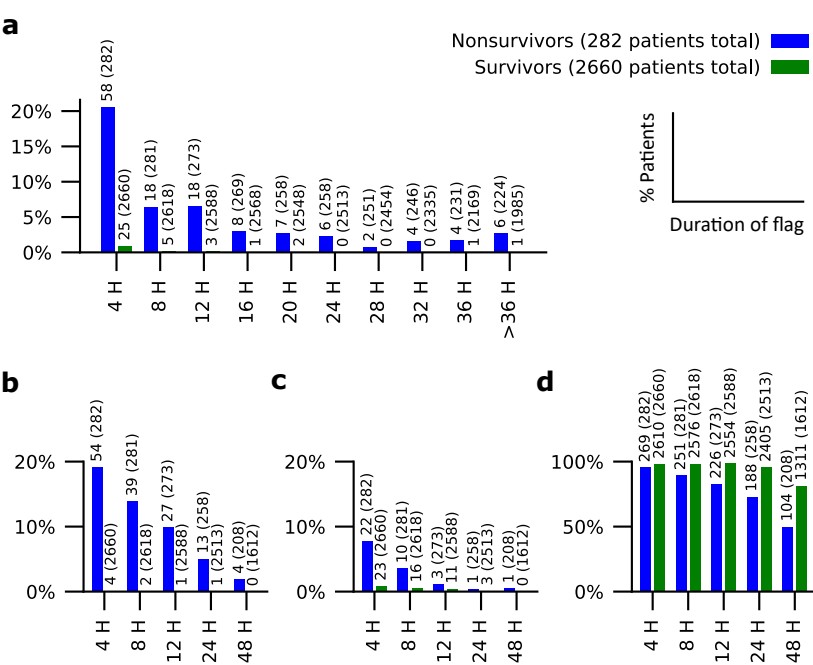

Fig. A6: **Flag duration for ICU patients.** Remaining on *confirmed red-flag* is measured for both survivor and nonsurvivor patients. **a** The bars represent the percentage of patients who experience at least one red-flag with the exact duration on the horizontal axis. Texts depict number of patients (out of total patients with the minimum of specified stay duration). **b** and **c** depict patients who "finish" their ICU stay remaining on red and yellow flags, respectively, at the final X hours before terminal. **d** presents patients who "start" their trajectory with *no flag* at all for the first X hours on the horizontal axis. We found that for the large part, both survivors and nonsurvivors start their trajectory without any flag, suggesting that they do not necessarily start with an unrecoverable situation. Further, nearly zero percent of survivors would raise and remain on red-flag for more than eight hours (even eight hours is quite rare compared to the total number of survivor patients). In contrast, nonsurvivor patients demonstrate a fat tail in the duration distribution **a** and repeatedly remain on the red-flag for eight hours or more. This result suggests that remaining on the red-flag for long periods strongly correlates with mortality, which is inline with our theoretical analysis.

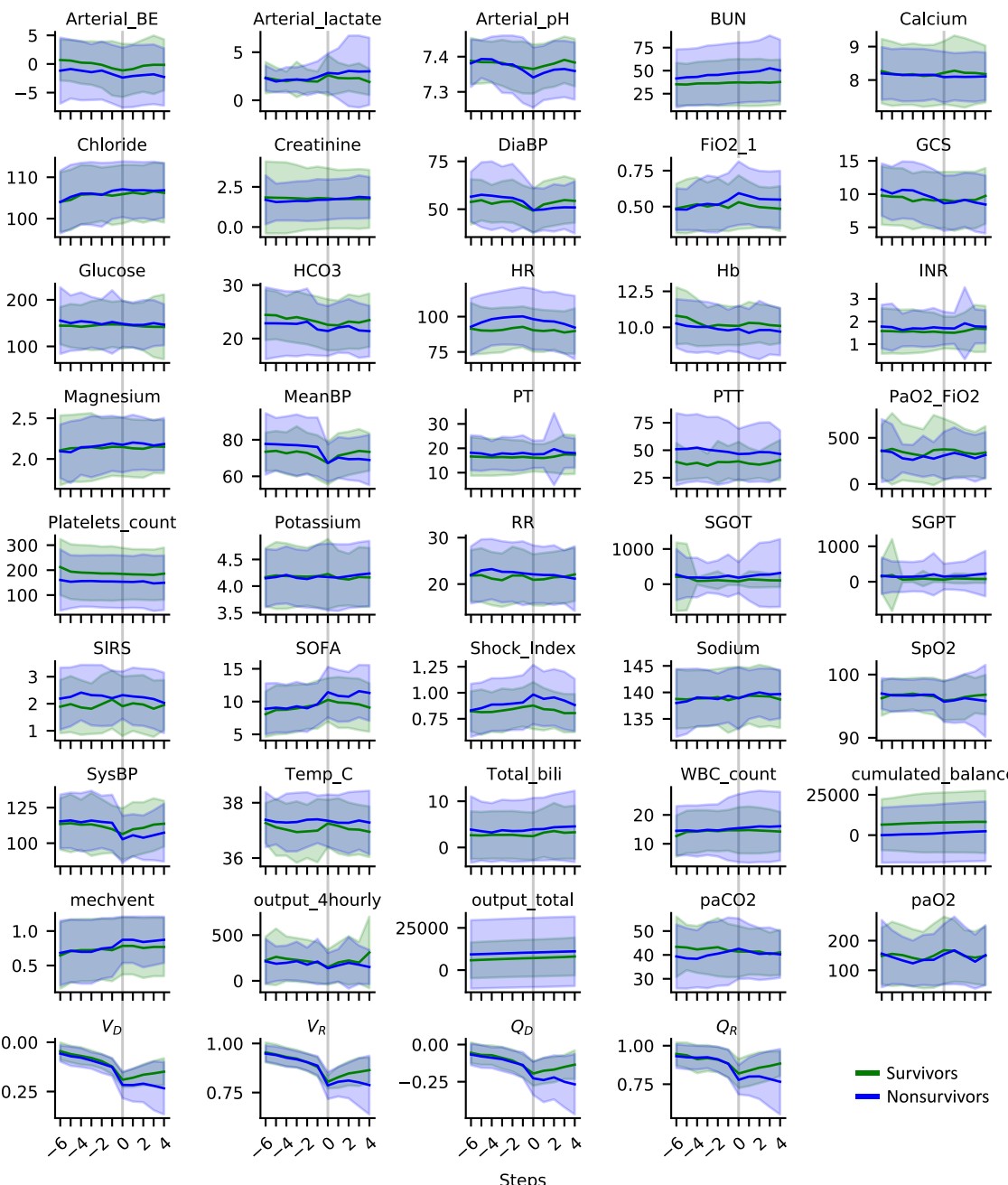

Fig. A7: **Signals prior to the first flag.** Complete list of vitals and standard measures in addition to our dead-end and secure values are shown for both survivor and nonsurvivor patients 24 hours (6 steps, 4 hours each) before and 16 hours (4 steps) after the first raised flag (red or yellow), indicated at point zero. Shaded areas represent standard deviation.

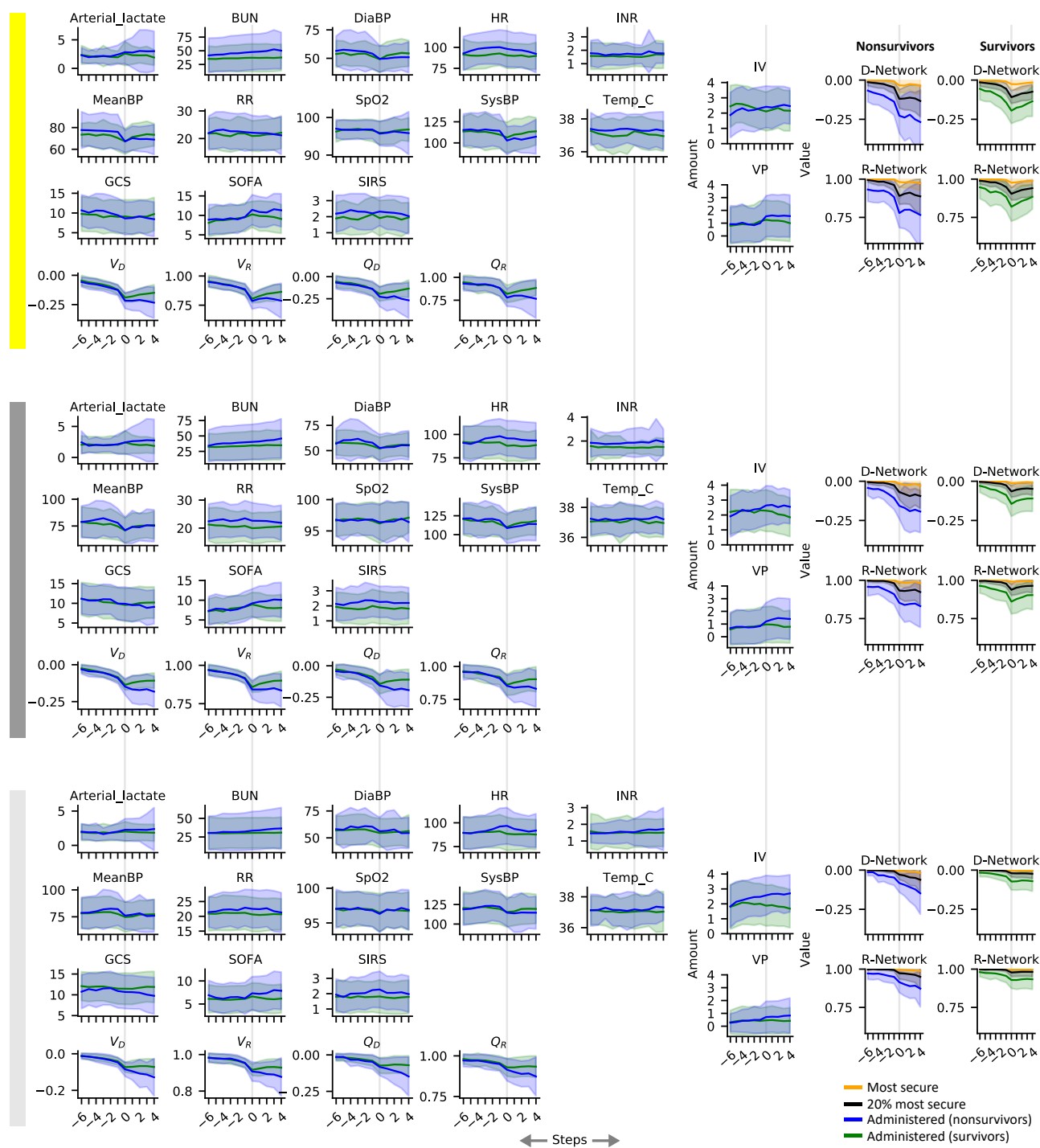

Fig. A8: **Trend of various measures before and after the first raised flags.** Various measures are shown 24 hours (6 steps) before and 16 hours (4 steps) after the first threshold crossing. The colors respectively corresponds to the following thresholds: yellow: $\delta_D = -0.15$, $\delta_R = 0.85$; dark grey: $\delta_D = -0.10$, $\delta_R = 0.90$; light grey: $\delta_D = -0.05$, $\delta_R = 0.95$. Shaded areas represent standard deviation.

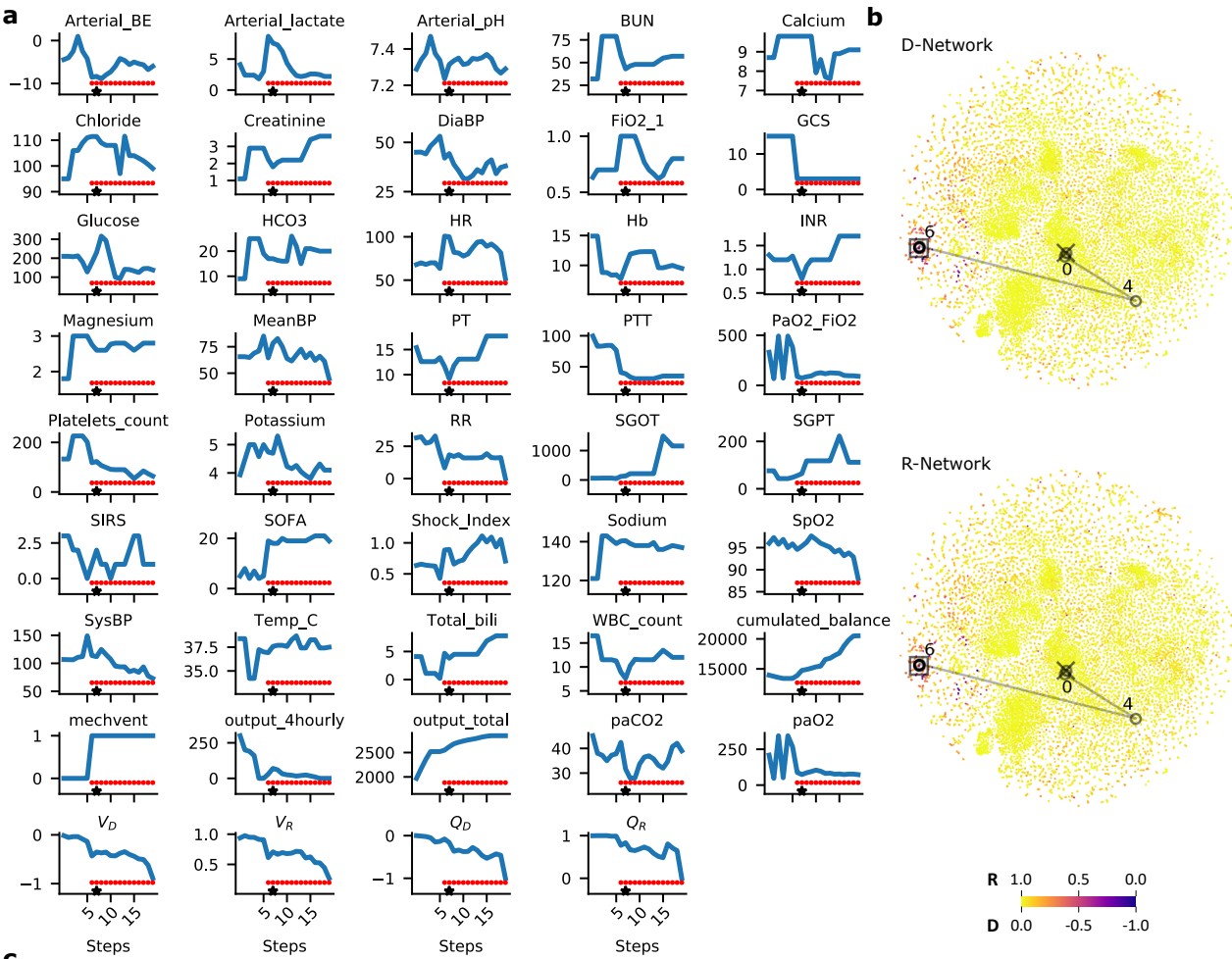

**c**

**Step 0: 2181-06-16 18:49:00 (Chest X-Ray Report):** "…mulitfocal pneumonia, asymmetric pulmonary edema, or both… Concern for fluid overload… worsening bilateral opacification…"

**Step 2: 2181-06-17 03:24:00 (Nursing Report):** "…[family] wish expressed that pt supported fully, including intubation if necessary,… pt is lethargic, answers questions intermittently w/ unclear speech… continue sepsis protocol…"

**Step 3: 2181-06-17 06:04:00 (Chest X-Ray Report):** "…Improved aeration of the lungs with features of fluid overload and possible worsening right effusion…"

**Step 4: 2181-06-17 12:23:00:** Patient intubated

**Step 4: 2181-06-17 13:05:00 (Chest X-Ray Report):** "Some worsening of airspace findings bilaterally in the lower lung zones -- fluid overload likely -- …"

**Step 6: 2181-06-17 18:07:00 (Nursing Report):** "Pt. intubated for impending resp. failure… Became hypotensive shortly after intubation and started on vasopressin… Lactate trending up… Awaiting brother's visit tonight to ? make cmo…"

**Step 8: 2181-06-18 03:37:00 (Nursing Report):** "Awaiting arrival of brother and continuing w/ full aggressive treatments until his arrival…"

**Step 9: 2181-06-18 05:13:00 (Nursing Report):** "…plan is to continue with support…"

**Step 11: 2181-06-18 16:36:00 (Nursing Report):** "Pt remains unresponsive, no longer breathing over vent. Lactate has been trending down… Brother has been bedside, is leaning toward CMO, will consult w/ other family… Continue current care…"

**Step 12: 2181-06-18 17:22:00 (Nursing Report):** "Brother arrived with sister… verbalizing wishes to withdraw life support, maintain comfort care…"

**Step 15: 2181-06-19 05:53:00 (Chest X-Ray Report):** "Multifocal infection including nodules in the left lower lung… Distension of the stomach with air and fluid is improved…"

**Step 15: 2181-06-19 06:16:00 (Nursing Report):** "Pt. continues to be non-responsive… Pt. made DNR… Continue current level of care, ? making pt. CMO if no improvement over next 24 hours…"

**Step 17: 2181-06-19 16:13:00 (Nursing Report):** "Pt. remains intubated… Dropped RR from 26 to 20… Family still undecided on whether to change pt's code status… Pt. remains unresponsive… Pt. is DNR, family meeting later to discuss status change…"

Fig. A9: **Complete analysis of nonsurvivor patient 262011. a** all the vitals, standard measures, max treatments, and network values for a nonsurvivor patient ICU-Stay-ID 262011. Red dots, yellow dots, and the asterisks show red and yellow flags and the presumed onset of sepsis, respectively. **b** patient's trajectory on the t-SNE plot, and **c** extracted chart notes from different source with their corresponding time stamp and quantized step.

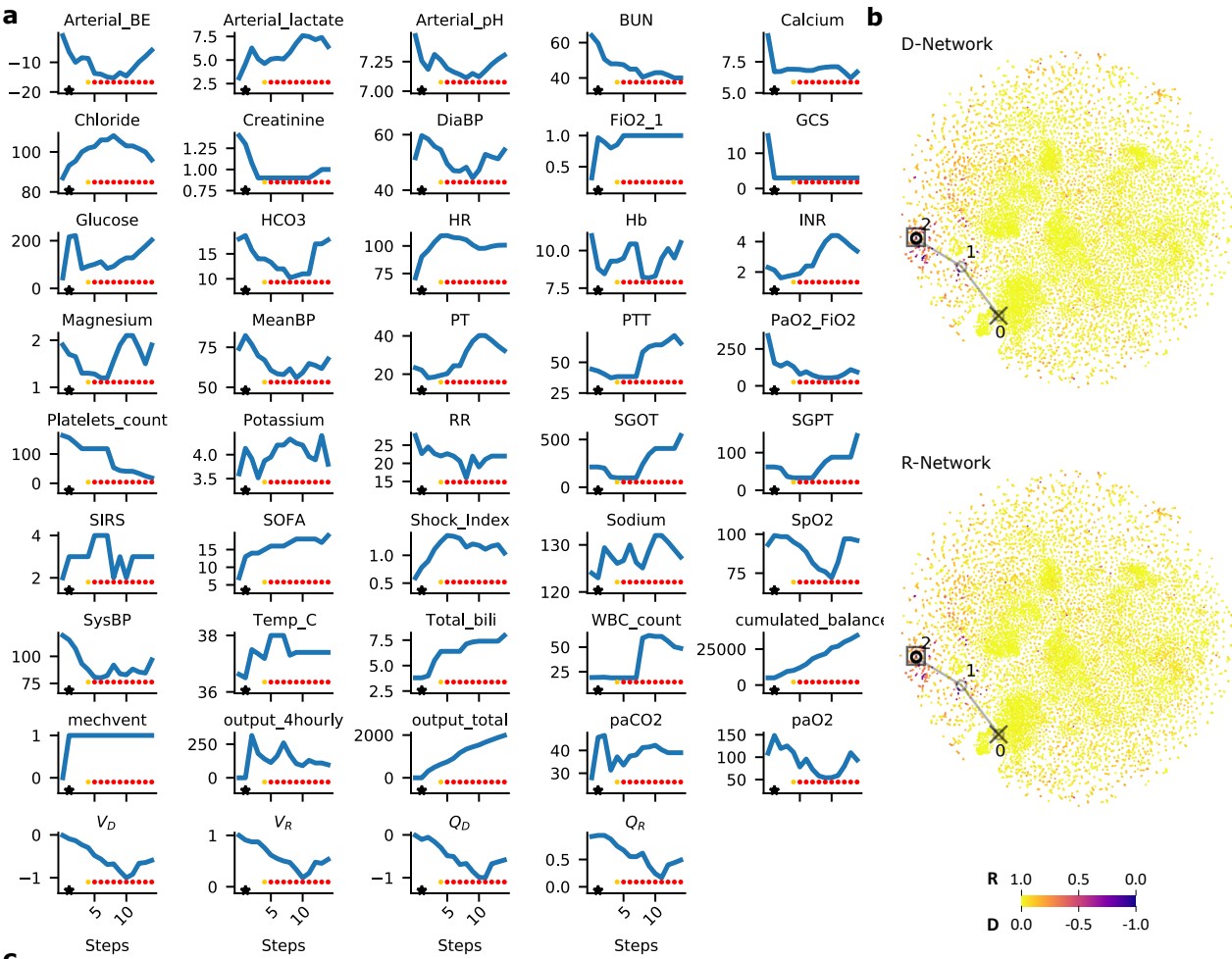

**c**

**Step 0: 2176-04-09 18:42:00 (Abdomen Report):** "Large abd wall hernia…p/w abd wall infection, likely necrotic. Also w/ diffuse TTP over abd… Visualized lung bases are clear"
**Step 0: 2176-04-09 20:48:00 (Chest port line placement):** "Abd hernia, now w/ hypoxia… w/ minimal linear opacity at the rt lung base"
**Step 0: 2176-04-09 20:48:00 (Chest port line placement):** "Atelectasis or developing infiltrate at base of rt lower lobe is less conspicuous"
**Step 1: 2176-04-09 21:20:00 (Abdomen Report; post intubation):** "Collapse of rt middle lobe, rt lower lobe and significant left to right cardiomediastinal shift…"
**Step 2: 2176-04-10 01:21:00 (Abdominal CT):** "Persistent volume loss in rt lunch w/ some expansion in rt middle lobe… Probably newly developing rt small pleural effusion"
**Step 3: 2176-04-10 05:09:00 (Nursing Report):** "Pt had hernia w/ necrotic abd wall… Pt has coarse bilateral LS w/ very thick brown secretions…Pt remains sedated w/ no spont. Respirations."
**Step 3: 2176-04-10 05:36:00 (Nursing Report):** "ABG's becoming progressively more acidotic…"
**Step 4: 2176-04-10 09:22:00 (Chest CT):** "Substantial clearing of opacification at the right base, consistent with re-expansion of lung following removal of mucous plug or repositioning of endotracheal tube…"
**Step 5: 2176-04-10 13:46:00 (Chest CT):** "Improved expansion or rt lower lobe… Minimal residual atelectasis is seen in rt middle lobe…"
**Step 6: 2176-04-10 17:31:00 (Nursing Report):** "Large volume resuscitation for hypotension… Persistently hypotensive… Lungs coarse, decreased at bases… DP/PT pulse present in AM, now absent… Code status changed to DNR…"
**Step 8: 2176-04-11 04:28:00 (Nursing Report):** "Pt on AC vent… Situation went from bad to worse…"
**Step 9: 2176-04-11 06:37:00 (Nursing Report):** "Pt sedated and paralyzed, doesn't appear to be in pain… Severe metabolic acidosis on max vaso and vent support… Worsening condition…"
**Step 11: 2176-04-11 16:19:00 (Nursing Report):** "Pt. remains intubated and currently vented on full support… pt. Remains metabolically acidotic and severely hypoxic…"
**Step 11: 2176-04-11 16:21:00 (Nursing Note):** "Pt. very fluid positive w/ total body anasarca. Areas of necrosis remaining… Very poor prognosis and condition… Pt remains critically ill w/ profound hypoxia and met. Acidosis as well as sepsis… Continue aggressive ICU care."
**Step 14: 2176-04-12 06:12:00 (Nursing report):** "Pt requiring multiple fluid boluses to maintain BP… Improved metabolic acidosis… on max vaso and vent support…"
[8 hours after this report, the family requests the patient to be made CMO and expires shortly thereafter]

Fig. A10: **Complete analysis of nonsurvivor patient 270174. a** all the vitals, standard measures, max treatments, and network values for a nonsurvivor patient ICU-Stay-ID 270174. Red dots, yellow dots, and the asterisks show red and yellow flags and the presumed onset of sepsis, respectively. **b** patient's trajectory on the t-SNE plot, and **c** extracted chart notes from different source with their corresponding time stamp and quantized step.

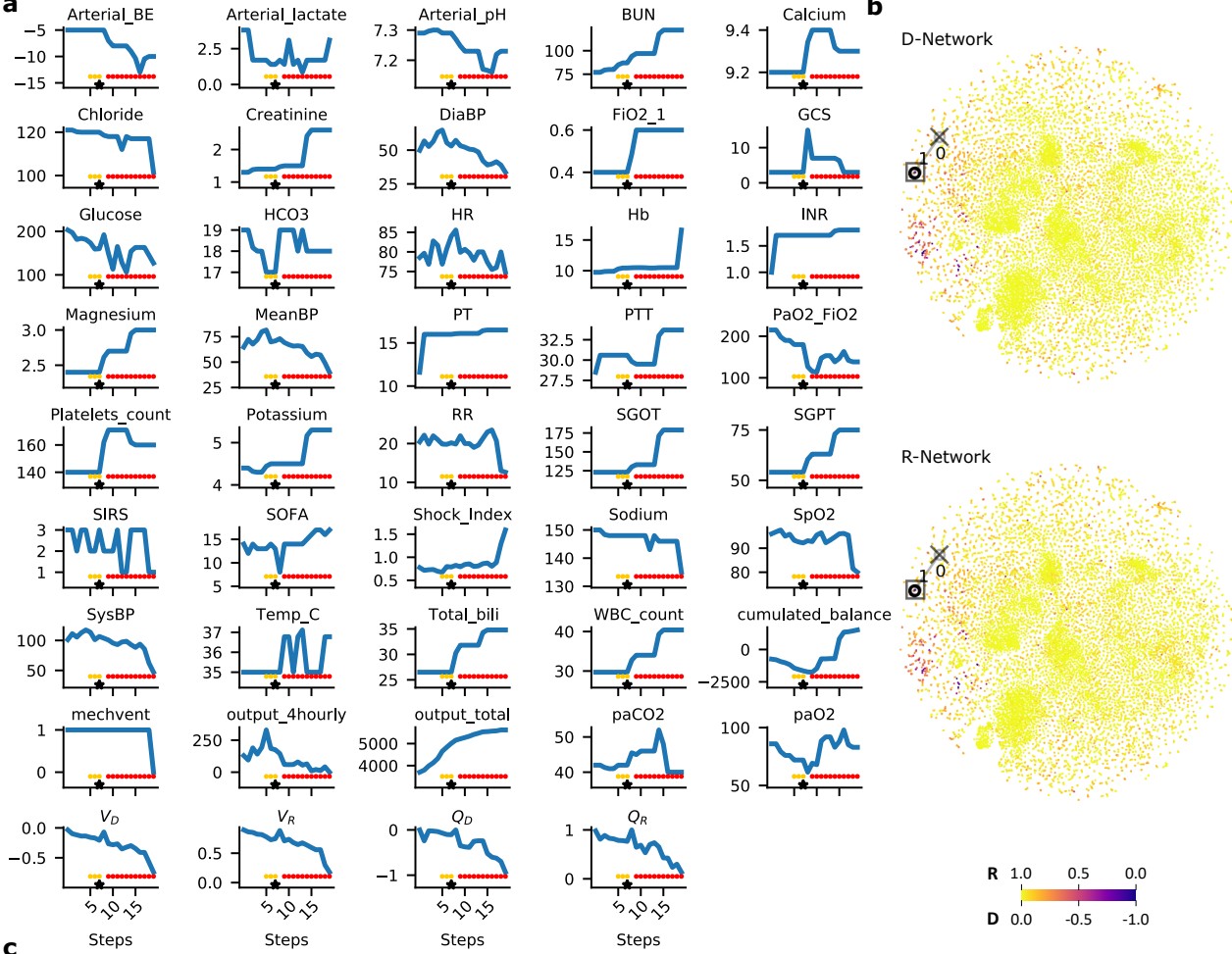

**c**

**Step 0:  2193-05-15 21:31:00 (CT Head Report):** "Liver failure... Uneven but reactive pupils on phys. exam..."
**Step 2:  2193-05-16 04:44:00 (Nursing Report):** "Pt. unresponsive to painful stimulation, extremities flaccid..."
**Step 2:  2193-05-16 05:44:00 (Chest X-Ray Report):** "Hepatic failure and GI bleed... Pt. remains intubated... Marked improvement of left sided pleural effusion..."
**Step 4:  2193-05-16 14:24:00 (Abdomen CT Report):** "Rising amylase, investigate for necrotizing pancreatitis... cirrhotic liver... opacification in the lower left lobe... suggesting a focal infectious or inflammatory process..."
**Step 5:  2193-05-16 15:25:00 (Nursing Report):** "Oxygenation improved!!! on R+L side w/ sat of 95-98%... In the setting of pancreatitis and worsening LFTs... family contacted by phone, informed of status and DNR status obtained..."
**Step 5:  2193-05-16 17:45:00 (Nursing Report):** "Pt. essentially unresponsive on fetanyl and ativan drips... He is overbreathing the vent..."
**Step 8:  2193-05-17 04:54:00 (Nursing Report):** "Plan family meeting..."
**Step 11:  2193-05-17 18:21:00 (Nursing Report):** "There were a few vent changes made in hopes of forcing a compensation... Pt. w/o any improvements, worsening acidosis..."
**Step 14:  2193-05-18 05:59:00 (Nursing Report):** "No spontaneous or purposeful movement... Worsening renal failure, worsening overall system failure..."
**Step 17:  2193-05-18 17:49:00 (Nursing Report):** "Pt. had issues w/ hypotension today... Continues to be acidotic... Family was called this AM and was told the severity of the situation... Pt. will be extubated and made CMO..."

Fig. A11: **Complete analysis of nonsurvivor patient 235403. a** all the vitals, standard measures, max treatments, and network values for a nonsurvivor patient ICU-Stay-ID 235403. Red dots, yellow dots, and the asterisks show red and yellow flags and the presumed onset of sepsis, respectively. **b** patient's trajectory on the t-SNE plot, and **c** extracted chart notes from different source with their corresponding time stamp and quantized step.