# OpenReview forum: "Medical Dead-ends and Learning to Identify High-Risk States and Treatments"
_NeurIPS.cc/2021/Conference — NeurIPS 2021 Poster_

### Official Review · Reviewer_iwUq · 2021-07-08

**Rating:** 7
**Confidence:** 4

**Summary:**

This paper studies a unique offline RL problem in the clinical domain -- discovering "dead-ends" -- which refers to identifying states from where any policy (even π*) will lead to eventual failure, and thereby identifying actions to avoid so that the system does not enter dead-ends.

With a specific MDP reward construction, the paper provides theoretical analysis shows a connection between value functions and the maximum/minimum probability of eventual success/failure (given a state or state-action pair), based on which thresholding can be applied to identify viable policies (and accordingly, actions that should be avoided) to adhere to an established definition of security.

The method is then applied to a toy domain as well as sepsis management in MIMIC-III data and showed some interesting and potentially useful results.

**Limitations And Societal Impact:**

**Limitations**
- ...were not explicitly discussed
- Despite using median Q-value to define the value function (L266), this doesn't avoid the problem of overestimation in offline RL.
- In addition, though formulated as not a typical RL problem where the goal is to learn optimal policies, the proposed approach still relies on estimating optimal value functions from offline data, and will suffer from similar issues of offline RL; the authors should mention and acknowledge such issues: confounding that are unaccounted for, limited exploration,

**Societal Impact**
- Should mention that the tools developed in this paper, and AI systems in general, are not meant to replace clinicians, but rather to assist and augment their decision making.

**Main Review:**

**Originality**: The main contribution of this work provides a novel interpretation of standard RL value functions, and in my opinion is valuable and inspiring.

**Quality**: Exposition is logical and informative. The main text summarized the intuitions and takeaways for the theory, while detailed proofs are provided in the appendix. I also appreciated the code release statement to facilitate reproducibility and future work.

**Clarity**: Overall this paper is nicely motivated and very clearly written. Evidently, the authors have made an effort to create visually appealing figures and illustrations.

**Significance**: Though motivated from a healthcare application, the proposed technique could be broadly relevant to RL where safety is critical.

---
**Detailed questions**
- L120: "In an offline setting with limited and non-exploratory data, inducing an optimal policy is not feasible [11]; hence, we do not specify the reward function of $M$." This sentence as written is a little misleading, because later you do specify the reward function (L148), and in fact, two reward functions for $M_D$ and $M_R$. I think what you mean is that you don't specify a reward function for which standard RL would optimize cumulative rewards, but in later sections present a specific design of reward (and discount factor) to assist in identifying dead-end/rescue states.
- L191 refers to Sec 4 for "practically select the thresholds $\delta_D$ and $\delta_R$" but I think the discussion appeared in Sec 5
- L218: in the toy example, $\delta_R$ threshold seems incorrect. Here since no dead-end should violate $\delta_R$ I think a valid threshold is just $0$. Also you might want to grey out / cross out the non-reachable states (barriers, life gates, death gates) in the heatmap so those 0 values from initialization are not conflated with the actual estimated 0 values.
- There are some interesting parallels between this work and Irpan et al., 2019 who also focused on goal-oriented MDP with sparse binary rewards. Their notion of "feasible" vs "catastrophic" is very similar to the "rescue" vs "dead-end" presented in this work. Although they focused on a different problem of off-policy evaluation, I believe it is beneficial to discuss how the two are similar, and most importantly, highlight the differences. For example, at a high level, "feasible" and "catastrophic" in Irpan et al. are asymmetrical with P(success|feasible)>0 under π*, P(success|catastrophic)=0 under π*, whereas "rescue" vs "dead-end" has a different asymmetry with P(recovery|rescue)=1 for some π, P(death|dead-end)=1 for all π; this work focuses on discovering critical states while Irpan et al. looked at classifying all state-action pairs. It also appears this work solves the problem using a different technique by casting the problem into two auxiliary MDPs (Irpan et al. used positive-unlabeled classification). [Alexander Irpan, Kanishka Rao, Konstantinos Bousmalis, Chris Harris, Julian Ibarz, Sergey Levine. "Off-Policy Evaluation via Off-Policy Classification". NeurIPS 2019. https://arxiv.org/abs/1906.01624 ]
- On the sepsis problem (and in general), since you mentioned and compared to GCS and SOFA, I think you need to highlight how this dead-end identification technique differs from a supervised sepsis mortality prediction model. One could design a flagging system by thresholding predicted scores. Perhaps reiterate the fact that V*/Q* functions implicitly anticipates the best treatments (the "*" - optimal policy) possible in the future, whereas a model trained via supervised learning can only predict the average outcome following the behavior policy.

**Time Spent Reviewing:**

4

---

> ### Author Response · Authors · 2021-08-10
> **Author Response**
>
> We thank the reviewer for the editorial comments and will consider them all. The various clarifications and detailed comments that have been posed here have helped to make the paper more clear and all-around better. We have attempted to respond to the most pressing comments below.
>
>
> ``Toy example:``
> The thresholds are correct. Just to clarify, $\delta_R$ separates rescue states from non-rescue ones, i.e., it splits the R values in such a way that $Q_R > \delta_R$ means the state-action pair *is* a rescue. The fact that it can be used as a *rescue test* to confirm dead-end values is then a consequence. The extreme case, which happened in the yellow states of this example, happens when encountering dead-end states whose (optimal) R values must be zero, affirming that there cannot be a way of reaching recovery. Remark that the rescue check (probing whether $Q_R < \delta_R$) still works correctly, as expected ($0 < \delta_R = 0.7$) -> if it had not held, then it would have signaled a possible false positive.
>
> We are additionally grateful for the suggestions to improve our visualization. We will certainly make these changes for the final, camera-ready version.
>
> ``interesting parallels between this work and Irpan et al., 2019``
>
> We thank the reviewer for bringing this paper to our attention and the clear explanation. We obviously missed this paper in our literature review and will properly cite this work. In a nutshell, while there are partial parallels in terms of grounding ideas, our theoretical development vastly diverges from Irpan et al., which is largely experimental and is centered wholly on policy evaluation rather than the assessment of specific decisions an agent may make. In application, our findings clearly deviate; we summarize key important differences as follows:
> 1) Their concept of *feasible* is simply being *non-catastrophic* and is different from *rescue*, which is a state where recovery is reachable w.p.1. (i.e., there is no adequate parallel for rescue states).
> 2) The properties of the Q function and how it formally links to the probabilities is not derived, discussed, or used in their work.
> 3) Their OPC metric is a proxy for evaluation/ranking learned policies. They do not use the framing to identify problematic or high-risk actions that may lead to catastrophic behavior. More accurately, there is no particular parallel for the concept of (treatment) security, its definition, and the formal guarantees which then shape the foundation of DeD.
> 4) In their work, the classification component is used to identify the value of state-action pairs on a binary {0, 1} scale. This makes negative behavior somewhat unidentifiable (they acknowledge this) from intermediate feasible states that do not correspond to terminal conditions.
> 5) Our dead-end construction (reward of -1 for bad outcomes + no-discounting) provides an inherently different value function, which (with a negative sign) formally gives rise to the minimum probability of bad outcomes in the future.
> 6) Side note: in dangerous and stochastic environments and for sufficiently long episodes, their Theorem 1 results in the trivial bound of $R(\pi)\ge 0$ (since the lower-bound becomes a negative value). Their experiments are restricted to robotic tasks and the Atari game of Pong; thus, this core problem has remained hidden in their work.
>
> ``On the sepsis problem (and in general), since you mentioned and compared to GCS and SOFA, I think you need to highlight how this dead-end identification technique differs from a supervised sepsis mortality prediction model...``
>
> A part that will be added to the paper will read as follows:
>
> >Value functions, by definition, encompass long-term consequences and are not myopic to possible immediate events, as opposed to supervised learning from immediate observation of an outcome. This inherent characteristic of value functions indeed yields the theoretical result presented by Lemma 2 (Appendix Sec. A1), one result of which is that $-Q_D$ corresponds to the minimum probability of a negative *future outcome*. Supervised learning from immediate outcomes, on the other hand, lacks this formal property; hence, it is not expected to provide parallel results with DeD.
>
> As we mention in our response to Reviewer vbiq with regards to supervised prediction models:
>
> Mortality prediction is estimating something different and potentially less useful to the clinicians (see *Maley, et al; reference below*). Where we focus on understanding the interaction between a treatment and patient state, and how that may eventually lead to mortality, mortality prediction is often done without consideration of treatment choice and the possibility of state modification. We aim to enable a more tactile interaction between attending clinicians and the algorithm, where in-context high-risk treatments are flagged as a form of cautionary recommendation.
>
> However, the proposed supervised prediction baseline seems like a reasonable approach to test and compare to the value functions learned through DeD. As mentioned in our response (2/2) to Reviewer AXB7, we are designing a silent, small observational trial with our clinical collaborators to help further validate the utility of DeD within clinical decision making. We will certainly include this suggested approach as an algorithmic alternative and test its comparative utility to DeD.
>
> *Maley, Jason H., et al. "Mortality prediction models, causal effects, and end-of-life decision making in the intensive care unit." BMJ Health & Care Informatics 27.3 (2020).*
>
> ``Limitations and societal impact:``
>
> We are grateful for the suggestions on elucidating the limitations of the algorithmic approaches underlying our proposed DeD method. Following the recommendations provided, we have clarified our paper accordingly. We regret that we were similarly less direct on possible societal impacts of using this algorithm in practice. As mentioned in our responses to Reviewers 5pYT and AXB7, we will include the following paragraphs in our paper to highlight our intentions with DeD and potential effects of misuse:
>
> >This work, or derivatives of it, should never be used to exclude patients from being treated, e.g., not admitting patients based on dead-end analysis. The treatment-avoidance part of our proposed approach is meant to shrink the scope of possible treatment options, and help the doctors to make better decisions. The flags that our approach, DeD, supplies are directly tied to specific treatment options rather than a binary treat/don’t treat decision. The intention of this approach is to assist clinicians by highlighting possibly unanticipated risks when making decisions and is not to be used as a stand-alone tool nor as a replacement of a human expert. Potential misuse of this algorithmic solution would carry significant risk to the well-being and survival of patients placed in a clinician’s care.
>
> >The primary goal of our work is to establish a proof of concept where especially high-risk treatments can be avoided, where possible, in context of a patient’s health condition. In acute care scenarios such as for patients with sepsis, treatments come with inherent risk profiles and potential harms. In these settings the tendency has become to overtreat patients with every attempt to ensure their survival, increasing the chance of clinical errors to occur [1]. Recent clinical research has sought to simplify practice to only the most necessary treatments [2]. In this spirit, we seek to assist clinicians by inferring the long-term impact of each available treatment in view of their risk of pushing the patient into an irrecoverable negative terminal condition.
>
> (Reference enumeration is in the context of the above paragraph)
>
> *[1] Carroll, Aaron E. "The high costs of unnecessary care." Jama 318.18 (2017): 1748-1749.*
>
> *[2] See: ​​https://www.bmj.com/too-much-medicine and http://jamanetwork.com/collection.aspx?categoryid=6017*

---

> > ### Comment · Reviewer_iwUq · 2021-08-22
> > **A few more comments/thoughts**
> >
> > Thank you authors for your response to my questions and to others'. I have a few further comments/suggestions:
> >
> > 1. In my opinion, some more edits to the introduction is needed to better position this work in relation to offline RL. For your main contributions, I think you need to clearly state the following two points: (1) paradigm shift from identifying the single best treatment to use, to identifying multiple treatments to avoid (if any), and (2) a reward design such that V/Q convey special meanings. Right now the main body of the introduction is focused on motivating and explaining (1), but (2) is not mentioned yet is actually the core technical contribution and all the theory of this paper. By stating (2) explicitly in the intro the readers should no longer have questions like "Is it possible to use other reward outcomes" by _Reviewer AXB7_. Similarly, consider emphasizing (1) more and clearly to address "No comparison to standard RL optimizing approaches or even just basic predictive models for death given inputs at each time point. How to know if this framework offers much advance or difference?" by _Reviewer vbiq_.
> >
> > 2. L82 you stated "expand theoretical results to offline settings". I don't believe your theory is necessarily specific to the offline setting, and this statement may incorrectly hint that your approach is a solution to offline RL, or you analyzed how this reward design works better than other rewards given limited exploration and sampling error in offline RL (which I don't think you did). Please consider adjusting this statement.
> >
> > 3. Even with this reward function your MIMIC sepsis experiment is still operating under offline RL to learn value functions, which still comes with the normal challenges of offline RL. I suggest you explicitly stating this as a limitation.
> >
> > 4. Can you perhaps list a few aspects you did not explore in this paper (e.g. quote _Reviewer vbiq_ "where it works and where it does not"), in the limitations section and to elucidate future research on the concept of RL dead-ends.

---

> > > ### Author Response · Authors · 2021-08-23
> > > **Thank you**
> > >
> > > We appreciate your continued constructive comments. These will certainly help us improve the paper's camera-ready version.
> > >
> > > `1.` All the items you mentioned will indeed help to better clarify our contributions. We will update the Introduction as suggested.
> > >
> > > `2.` We agree. We should also note that the security condition (and DeD as the method to formally enable it) is crucial in the *absence* of an optimal policy, however it arises. This is always the case in offline scenarios with severe data limitations (hence, finding an optimal policy is impossible). Nevertheless, this may also happen in online scenarios during the exploration phase or if the behavioral policy is unable to cover the entirety of the state space for any reason. The online exploration case is indeed the subject of [Fatemi et al.], which we explain in our Related Work section. We will clarify the statement accordingly.
> > >
> > > `3.` We explained this point in detail to Reviewer AXB7. Even with limited data, the offline problem of DeD can be exponentially more reliable than the original control problem (finding an optimal policy to suggest the best treatment). This has been proved through Proposition 1 in the Appendix. Having said that, it is certainly correct that the induced value functions are not fully accurate; hence, errors are inevitable. We discuss in the paper how to mitigate the error by finding good thresholds. However, we agree that this is a limitation (just as in any AI method) and we will make this explicit in the conclusion section.
> > >
> > > `4.` There are a few points that may be noted as the limitations of the current submission. We should emphasize that attending to these points are beyond the scope of this paper, yet these are first-hand directions for future research. Several of these are being integrated into our follow-on study looking at developing a soft-trial of DeD with our clinical collaborators.
> > >
> > > **4.1.)** Investigation of how a learned DeD model is sensitive to demographic information; hence, how demographic bias can influence the final decision-making point both in terms of treatment avoidance, and call for emergency.
> > >
> > > **4.2.)** Sensitivity analysis of DeD with respect to various EHR variables as well as the patient's initial state.
> > >
> > > **4.3.)** Using contemporary Offline RL methods for value function estimation and comparing their results to see which fits best for the specifications of DeD, particularly that gamma is one [as explicitly mentioned, we have used DDQN to keep the implementation simple. We are preparing our codebase for public release, it should be quite straightforward to try other RL techniques within the DeD framework].
> > >
> > > **4.4.)** External validation of DeD results. This may be exercised at different levels:
> > >
> > >     A.) The R-Network can be trained using data from a different hospital, hence the rescue-test becomes fully independent, which makes DeD results becoming more authentic. Note that the D- and R-Networks are trained separately and independently; thereby, they can be trained on different data sources.
> > >
> > >     B.) After DeD modules are fully trained, they can be tested on data sourced from a different hospital (similar to what has been done in [Komorowski et al.]), and
> > >
> > >     C.) human physicians can investigate and validate the results, particularly in the case of treatment avoidance.

---

### Official Review · Reviewer_vbiq · 2021-07-16

**Rating:** 6
**Confidence:** 4

**Summary:**

Offline reinforcement learning approach which, instead of trying to achieve optimal outcomes, learns to instead probabilistically avoid bad outcomes ("dead end" states that will inevitably lead to a bad outcome state = death). Demonstrated and evaluated against the MIMIC3 sepsis dataset with 5*5 treatment / action choices at each state in terms of IVFluids and vasopressor dosing.


**Ethical Concerns:**

No major ethical concerns, though not specifically addressed in primary manuscript.
Conceivably if such algorithms were implemented into automated care / medication titration, there could be high stakes ethical issues of reliability and consistency.
Does appropriately bring up the likely intractable ethical limitation of doing full reinforcement learning on medical care, which would require exploring non-standard / off-policy treatment actions with potentially adverse outcomes.


**Limitations And Societal Impact:**

Prevalent and open benchmark dataset that many can compare against, but with inevitable retrospective data limitations and missingness. E.g., sepsis outcomes are affected by many choices beyond IVFluid and vasopressor doses. Likely confounding by indication as those treatment/action choices may simply be indicators of severity of illness, not necessarily poor treatment choices. E.g., finding association with "comfort measures only" with bad outcomes doesn't make much sense if all model is doing is basically just restating what the clinicians have already assessed. Adding some concrete examples of treatment courses the model recommends may add some face validity (or identify the above types of issues to address).


**Main Review:**

Good introduction reviewing the limits in observational/offline learning with supervised prediction assumes optimal behavior and reinforcement learning really requires the ability to explore options that don't exist and may very well be unethical to test in real-world, high-stakes (healthcare) settings.

An interesting problem construction that abandons trying to find an optimal policy based on offline/retrospective data, instead predictive networks and latent state construction to probabilistically avoid dead end states that will lead to bad outcomes.
Important problem setup of offline RL being limited, but approach as described still doesn't really seem to address the fundamental (and likely intractable) problem eluded about limited data for offline RL in the absence of exhaustive search/exploration/experimentation options.
No comparison to standard RL optimizing approaches or even just basic predictive models for death given inputs at each time point. How to know if this framework offers much advance or difference?

Reasonable use of MIMIC dataset as training and evaluation platform, but would benefit from more description of the 47 observation variables used.

Minor: Life-gate example is only superficially described with limited results to show, so not clear what point of contribution trying to make to the paper with it.


**Time Spent Reviewing:**

2

---

> ### Author Response · Authors · 2021-08-10
> **Author Response and Rebuttal**
>
> We appreciate the feedback that has been provided through this review and have done our best to respond and provide clarification to the concerns that were raised. We have tried to format our response such that a quote or comment from the review is highlighted with our response directly following. All additional references are italicized below the corresponding section.
>
> ``approach as described still doesn't really seem to address the fundamental (and likely intractable) problem eluded about limited data for offline RL in the absence of exhaustive search/exploration/experimentation options``
>
> In no way did we intend to give the impression that our proposed method, DeD, should be considered as a general solution to the major challenges facing offline RL. We do however claim (as stated on **lines 40-44, 62-64**) that DeD is a potentially useful tool when faced with “limited data” in safety-critical sequential decision-making situations.
>
> *As also responded to Reviewer AXB7:*
>
> 1) The proposed paradigm-shift indeed eliminates the requirement of having full exploration. Inferred from **Proposition 1** and **Remark 3** (**Appendix Sec. A1**), in general, learning dead-end values is an exponentially smaller problem compared to the original problem of finding the optimal control policy, with no need to learn the value of all state-action pairs. Hence, given the same data, it should be expected to achieve far more reliable values from learning $Q_D$ and $Q_R$ compared to the case of learning the optimal values for the sake of control, i.e., suggesting best actions.
>
> 2) Ideally, using optimistic learning, the worst that can happen are false negatives. This means that the agent would remain silent and leave decision-making completely to the clinicians (as a side note, the overestimation nature of RL also suppresses false positive cases). We emphasize that DeD has a passive nature by definition: Looking at the security condition, it is clear that the method formally engages only when the probability of danger gets high. Further, when there is no flag, it means either the state/treatment is not high-risk or enough data has not been available to learn from. Either case, the clinical decision-making would not be falsely invaded.
>
> ``No comparison to standard RL optimizing approaches or even just basic predictive models for death given inputs at each time point. How to know if this framework offers much advance or difference?``
>
> Due to severe data limitations, i.e., small sample issues in the vast number of potential combinations of treatments and patient states, no standard RL approach provides any reliable and even close to optimal control policy - please see the **Introduction** for more details and cited references. DeD is the first work to the best of our knowledge that shifts the problem to shrinking the scope of decision-making policy by refraining from high-risk actions; it further signals when the state becomes high-risk. There is no exact parallel for this work to be used as a baseline. On the other hand, we also see this paper as a first step to still make use of existing data, even with the knowledge that inducing an optimal control is not feasible in any capacity.
>
> In regards to the use of a mortality prediction model as a potential baseline, we note that mortality prediction is estimating something different and potentially less useful to the clinicians (*see Maley, et al; reference below*). Where we focus on understanding the interaction between a treatment and patient state, and how that may eventually lead to mortality. Mortality prediction is often done without consideration of treatment choice and the possibility of state modification as a consequence of that choice. Specifically, in a supervised learning problem where mortality is our outcome of interest and past treatments/states are our inputs, we want to establish the extended impact of a set of potential current actions rather than predicting mortality itself. Further, there is significantly less actionability in response to a prediction of mortality rather than a flag about a specific treatment. For these reasons, a supervised predictive model would not be an informative baseline.
>
> *Maley, Jason H., et al. "Mortality prediction models, causal effects, and end-of-life decision making in the intensive care unit." BMJ Health & Care Informatics 27.3 (2020).*
>
> ``would benefit from more description of the 47 observation variables used``
>
> We agree that a more thorough explanation of the features used in this study would be useful in the main-body of the paper. Due to space constraints, we relegated the specifics about the features and dataset construction to **Appendix A5** so as to not distract from our core technical contributions  (medical dead-ends and security, DeD methodology and formal results, and experimental findings). Included in that section is a table (see **Table A2**) that walks through summary statistics of each feature as well as provides some additional description.
>
> ``Life-gate example is only superficially described with limited results to show, so not clear what point of contribution trying to make to the paper with it.``
>
> The purpose of life-gate was to demonstrate, in a controllable fashion, the utility and validity of **Theorem 1** before it is used in a real-world scenario with many confounding factors and limitations. We will make this more clear in the initial sentences of **Section 3.6**. Due to space constraints we could not include an extensive description of the domain in the main body of the paper. We did however expand on this discussion and include a much more detailed description in **Appendix A2** (as noted at the end of the first paragraph of **Section 3.6**)

---

### Official Review · Reviewer_AXB7 · 2021-07-16

**Rating:** 6
**Confidence:** 3

**Summary:**

In this paper, the authors propose to adapt the dead-end analysis in reinforcement learning to study “medical dead-end” for patients in the intensive care units. They aim to identify the “dead-end” states where a patient will expire regardless of any future treatment sequence. “Non-secure” treatments may be identified based on their probability of leading to dead-ends. Theoretical and empirical analyses were conducted.

**Limitations And Societal Impact:**

The authors should discuss more on the limitations, and there is no discussion on societal impact. I think the lack of exploration may also affect the validity of dead-end identifications. And how generalizable is the method to future patients should be discussed especially considering the application to ICU patients.

**Main Review:**

I think bringing the dead-end analysis to the medical setting is very meaningful. It may give the clinicians more insights into patients’ states and designing treatment plans. The application setting is novel. But it may be better for the authors to highlight the main difference in methodology for the proposed method with the existing dead-end analysis and the challenges in the setting adaptation.

This paper is in general well structured, and various experiments were conducted with thoughtful interpretations. The “state construction” step is well explained but more details may be needed for the R- and D-network models. See below for more detailed comments.

Detailed comments:

It is true that the task of learning optimal behavior might fail due to a lack of exploration. But can the authors discuss how the lack of exploration would affect their dead-end analysis? For example, is it also possible that the identified dead-end states are not true dead-ends because some treatment sequences were not explored in the data?

The two Markov Decision processes D-network and R-network are built on deep Q-networks. It seems to me both the state transition probabilities and the conditional reward outcomes need to be learned. Is the objective function to minimize the error in estimating the states and rewards? It is not clear to me in supplementary A4, how the “second layer directly outputs 25 nodes corresponding to the 25 treatments” correspond to the learning objective. It would be helpful to provide more details on the model structure and learning algorithm.

Can the authors explain why the D-network and R-network seem to deteriorate much more aggressively than the observed clinical measures, for example in Figure 3?

For the D-network and R-network, a reward outcome is defined as -1/0 and 1/0. Is it possible to use other reward outcomes, e.g., outcome variables on the continuous scale which may carry more information? Will the method and theory still hold?

The state variables are assumed to be discrete in the paper. Can the authors discuss the rationale, and can the state variables be modeled as continuous?

It seems the threshold for defining the dead-ends plays an important role. Can the authors discuss how the thresholds were chosen? Any guidance or external validations for the thresholds?

In order to be implemented in the clinical setting, good validation is important. Can the authors discuss how the dead-end states can be validated in practice? Related to the previous question, the trend in the D-network and R-network seem to be more aggressive than the observed clinical measures. How to validate the estimated quantity with observed measures need to be further discussed.


**Time Spent Reviewing:**

6 hours

---

> ### Author Response · Authors · 2021-08-10
> **Author Response and Rebuttal**
>
> Thank you for the time that was spent reading and evaluating our work. We appreciate the feedback and have done our best to respond and provide clarification to the considerations raised in this review over 2 comments (due to space limitations). We have tried to format our response such that a quote or comment from the review is highlighted with our response directly following. All additional references are italicized below the corresponding section.
>
> ``can the authors discuss how the lack of exploration would affect their dead-end analysis?``
>
> 1) The proposed paradigm-shift indeed eliminates the requirement of having full exploration. Inferred from **Proposition 1** and **Remark 3** (**Appendix Sec. A1**), in general, learning dead-end values is an exponentially smaller problem compared to the original problem of finding the optimal control policy, with no need to learn the value of all state-action pairs. Hence, given the same data, it should be expected to achieve far more reliable values from learning $Q_D$ and $Q_R$ compared to the case of learning the optimal values for the sake of control (i.e., suggesting best actions).
>
> 2) Ideally, using optimistic learning, the worst that can happen are false negatives. This means that the agent would remain silent and leave decision-making completely to the clinicians (as a side note, the overestimation nature of RL also suppresses false positive cases). We emphasize that DeD has a passive nature by definition: Looking at the security condition, it is clear that the method formally engages only when the probability of danger gets high. Further, when there is no flag, it means either the state/treatment is not high-risk or enough data has not been available to learn from. Either case, the clinical decision-making would not be falsely invaded.
>
> ``"it may be better for the authors to highlight the main difference in methodology for the proposed method with the existing dead-end analysis and the challenges in the setting adaptation"``
>
> As discussed in **Section 2** (Related Work), the existing dead-end analysis solely concerns exploration in the online RL settings. Adopting the basic concepts and expanding the ideas to the offline case is totally novel. The main challenge in the offline mode is that learning the value functions would naturally involve systematic errors (simply due to data limitation and not the algorithm). We discuss in the paper that this is expected to be exponentially less harmful compared to learning optimal control policies. Further, using optimistic learning, the impact of lack of accuracy is largely limited to false negatives. Meaning that the worst case (to a large extent) is that the agent does not help clinicians and fully leaves decision-making to them.
>
> ``It seems to me both the state transition probabilities and the conditional reward outcomes need to be learned. Is the objective function to minimize the error in estimating the states and rewards?``
>
> By construction, D- and R-networks learn the (optimal) value function of the two MDP’s discussed in the paper (i.e., the original environment with the assigned rewards and no discounting). Both are legitimate RL problems and any model-free or model-based approach may be used to learn the corresponding value functions, $Q_D$ and $Q_R$. There is no particular necessity to use a reward model or learning the transition probabilities in any way. As detailed in the paper, we used double-DQN algorithm (which is model-free).
>
> ``details on D- and R- networks`` + ``It is not clear to me in supplementary A4, how the “second layer directly outputs 25 nodes corresponding to the 25 treatments” correspond to the learning objective. It would be helpful to provide more details on the model structure and learning algorithm.``
>
> Both $Q_D$ and $Q_R$ have the same architecture, already detailed in **Sections 4 and A4**, and are trained separately using their corresponding reward definition and the DDQN algorithm (*see van Hasselt, et al 2016*) with identical hyper parameters. Each neural net has one hidden layer and the second layer means their output layer (it might have been the source of confusion). Hence, the output layer of each neural net consists of 25 nonlinear, fully connected nodes, corresponding to the Q value of the 25 possible treatment options. The loss function and training process simply follow the standard DQN/DDQN implementations. As the construction and algorithm for training these networks is quite standard, we didn’t feel the need to provide extensive details. We see, in comparison to how we outlined the SC-Network in the Appendix, that there is some imbalance. We will work to even up the discussion; the code is also being prepared to be made publicly available.
>
> *Van Hasselt, Hado, Arthur Guez, and David Silver. "Deep reinforcement learning with double q-learning." Proceedings of the AAAI conference on artificial intelligence. Vol. 30. No. 1. 2016.*
>
> ``Can the authors explain why the D-network and R-network seem to deteriorate much more aggressively than the observed clinical measures, for example in Figure 3?``
>
> The value functions are indeed predictive of future events as a result of observing all the state components and not just SOFA or other single measures. Further, the state, in turn, also includes the patient’s history through the state-construction process. Hence, it should be expected that $Q_D$ and $Q_R$ provide significantly more information about the patient’s condition in terms of future events of mortality and recovery. In a related note, one may also see $Q_D$ and $Q_R$ as an approximate form of the Successor Representation (see Dayan 1993).
>
> *Dayan, Peter. "Improving generalization for temporal difference learning: The successor representation." Neural Computation 5.4 (1993): 613-624.*
>
> ``For the D-network and R-network, a reward outcome is defined as -1/0 and 1/0. Is it possible to use other reward outcomes, e.g., outcome variables on the continuous scale which may carry more information? Will the method and theory still hold?``
>
> The choice of reward functions and discount factor is dictated by our theory and is critical in establishing the connection between value functions and the security condition; they are not arbitrary. Formally speaking, this construction results in the value functions which precisely guarantee the security condition. As a mathematical observation, however, one can see that for example a reward in $(-1, 0]$ for $Q_D$ would soften the security condition (i.e., some actions might not be secure, but will not be eliminated by DeD if such a reward is being used during the training).
>
> The use of distinct, sharp boundaries to define undesirable outcomes is also well motivated by use cases within safety-critical domains, such as healthcare, which lie at the foundation of our work. Scenarios or tasks where the rewards cannot be shifted to match this setting (e.g. where intermediate rewards are informative and necessary) are a setting of future work.
>
> ``The state variables are assumed to be discrete in the paper. Can the authors discuss the rationale, and can the state variables be modeled as continuous?``
>
> Please see the **footnote on page 3**. The discrete state is used for simplicity of the presentation. All the theoretical results can be extended to continuous states by properly replacing summations with integrals (as it is common in the RL literature). Further, it is critical to point out that all experiments using real EHR data feature continuous state variables, with discrete actions.
>
> ``Can the authors discuss how the thresholds were chosen?``
>
> We found that a good way to select effective thresholds is to make a bar plot of $Q_D$ and $Q_R$ over different buckets along backward time steps: split each of the intervals of [-1, 0] and [0, 1] into say N buckets and look at the values of each bucket moving backward in time, i.e., start from the end of all trajectories and go M steps backward. The rationale behind this technique is that the last state is either death or recovery, hence the value should be at its largest magnitude and should decay as we step-by-step move away from the terminal point. A good threshold corresponds to the (nearly) largest bucket that best distinguishes survivors from non-survivors, using a validation set. For our case, this process is detailed in the **Appendix, Figure A5**.

---

> > ### Author Response · Authors · 2021-08-10
> > **Author Response (continued) 2/2**
> >
> > `` Can the authors discuss how the dead-end states can be validated in practice?``
> >
> > In terms how dead-end states can be validated using the analytical tools derived from the D- and R- networks learned through DeD, we provided trajectories of several non-survivor patients in the appendix (**Figures A9-12**) where we also provided t-SNE plots of the states (dimensionally reduced). Additionally, we included comments from actual chart notes of the patients with their corresponding time stamp. The state transition in particular depicts how patients experience a jump to areas in the t-SNE plot which seemingly belong to eventual mortality. Some non-survivors start with an initial state in those areas, meaning that they are already on an almost-deadend trajectory with little chance of recovery. We additionally intend to release code for similar examination of any arbitrary patient within the patient cohort we extracted from the MIMIC-III database.
> >
> > In terms of real validation in clinical practice, randomized controlled trials are the obvious gold standard to ensure that our treatment avoidance recommendations do not worsen outcomes in patients, but potentially improve them. There are other styles of small-scale observational settings that are possible, including mirror or siamese studies where two “similar” patients are treated by care teams that either have a tool or do not. There are other non-random factors that may come into play in any setting other than an RCT, but by doing smaller scale observational studies of this kind, we may be able to capture what impact these policy-based avoidance flags have on clinical introspection during decision making, aside from clinician's taking the time to inspect the validity of flagged states and treatments in a hypothetical care scenario. We are currently designing such observational trials with our clinical collaborators as a feature of our ongoing and future work.
> >
> >
> > `` societal impact:``
> >
> > In response to similar concerns raised by Reviewers 5pYT and iwUq, we include the statements we were not clear enough in the draft that was submitted:
> >
> > >This work, or derivatives of it, should ever be used to exclude patients from being treated, e.g., not admitting patients based on dead-end analysis. The treatment-avoidance part of our proposed approach is meant to shrink the scope of possible treatment options, and help the doctors to make better decisions. The flags that our approach, DeD, supplies are directly tied to specific treatment options rather than a binary treat/don’t treat decision. The intention of this approach is to assist clinicians by highlighting possibly unanticipated risks when making decisions and is not to be used as a stand-alone tool nor as a replacement of a human expert. Potential misuse of this algorithmic solution would carry significant risk to the well-being and survival of patients placed in a clinician’s care.
> >
> > >The primary goal of our work is to establish a proof of concept where especially high-risk treatments can be avoided, where possible, in context of a patient’s health condition. In acute care scenarios such as for patients with sepsis, treatments come with inherent risk profiles and potential harms. In these settings the tendency has become to overtreat patients with every attempt to ensure their survival, increasing the chance of clinical errors to occur [1]. Recent clinical research has sought to simplify practice to only the most necessary treatments [2]. In this spirit, we seek to assist clinicians by inferring the long-term impact of each available treatment in view of their risk of pushing the patient into an irrecoverable negative terminal condition.
> >
> > (Reference enumeration is in the context of the above paragraph)
> >
> > *[1] Carroll, Aaron E. "The high costs of unnecessary care." Jama 318.18 (2017): 1748-1749.*
> >
> > *[2] See: ​​https://www.bmj.com/too-much-medicine and http://jamanetwork.com/collection.aspx?categoryid=6017*

---

### Official Review · Reviewer_5pYT · 2021-07-20

**Rating:** 6
**Confidence:** 4

**Summary:**


This paper explores the problem of sequential decision-making in the medical domain, specifically in offline settings (also known as "logged" data) wherein the data available is insufficient and the logged policy is not optimal. In this setting, both supervised learning and policy identification via RL may not be viable since the former assumes the logged policy is optimal and the latter explores counterfactual alternatives that are not observed in the data.

The paper has both a **conceptual** contribution and an **algorithmic** one. Conceptually, the authors introduce two new notions:

- *Medical dead ends*: These are states (within the state-space) that the patient is inoperable, i.e. no  treatment sequence will be able to move them back to a more healthy state.
- *Treatment security*: the idea of avoiding treatments with probability proportional to their chance of leading to dead-ends, rather than optimize the treatments to maximize health outcomes.

From an algorithmic perspective, the authors train three independent deep neural models for automated state construction, dead-end discovery and confirmation. Experiments were conducted on the MIMIC data set to demonstrate these concepts and algorithms in practice.

**Ethics Review Area:**

["Inappropriate Potential Applications & Impact  (e.g., human rights concerns)"]

**Limitations And Societal Impact:**

The paper advocates a patient treatment approach that is focused on figuring out which treatments should be avoided, or equivalently which patients should *not* be treated. This is indeed a valid perspective to consider from a technical point of view, but because of the ethical dimension to this problem, the paper should include a discussion paragraph with caveats on this approach, its limitations and the contexts in which it would be sensible. An ethics review by a reviewer with a clinical background is worthwhile.

**Main Review:**


This is an interesting approach to ICU patient management that takes on a more realistic way for handling offline logged data, focusing on treatments to avoid rather than optimal treatments. For the most part, this paper is a direct application of the work on dead-end and secure exploration by Fatemi et al. in [44]; the technical concepts and modeling aspects of the paper are not particularly novel. However, the applications considered in the paper are important and the results are worthy of publication, especially the results in Section 5.2 (first flag analysis). This prompts me to give a **borderline acceptance** rating. This paper would normally go into the application track where the focus is on the impact of the proposed model in a given application domain.

**+ Comments and thoughts on the concept of dead-end states**

The proposed framework only makes sense in setups where patients' states are rapidly changing and some treatments may exacerbate progression to an inoperable state. In most clinical applications, some treatment may be ineffective for terminally-ill patients, but in most cases treatments will not put a patient in an irreversibly bad states. This setup may be very appropriate for sepsis but not very relevant to most diseases with sequential treatment decisions, where figuring out the "treatments to avoid" won't be very helpful.

**+ Comments on the experiments**

While the results of Section 5.2 are interesting, I wonder how Figure 3d conclusively indicates treatments cause entrance into a dead-end. In general, I am not sure if the model can really make these kinds of causal claims since patients' vitals or other unobserved factors may have prompted both the treatment and the dead-end. A more elaborate discussion of this would be helpful.


**Needs Ethics Review:**

Yes

**Time Spent Reviewing:**

5+ hours

---

> ### Author Response · Authors · 2021-08-10
> **Author Response and Rebuttal**
>
> First, we wanted to express our appreciation for the time that was spent reading and evaluating our work. We appreciate the feedback and have done our best to respond and provide clarification to the considerations raised in this review. We have tried to format our response such that a quote or comment from the review is highlighted with our response directly following. All additional references are italicized below the corresponding section.
>
>
> ``Comments and thoughts on the concept of dead-end states:``
>
> We agree with the reviewer. As also pointed out in the **Introduction**, the presented methodology is most appropriate in safety-critical domains, where *high-risk* states and/or actions exist. We should further re-emphasize that in addition to treatment-avoidance, DeD also flags high-risk states. Hence, other safety-critical applications, which involve high-risk states, but not necessarily dangerous actions can still benefit from this second aspect of DeD.
>
>
> ``Comments on experiments``
>
> In **Fig. 3**, a few points are important to note:
> 1) **Fig 3-d**: the *average* usage of both actions is quite similar for both survivors and non-survivors until around 8 hours prior to *the first flag*, where the course of treatment (both IV and VP) seems to diverge for the two groups. This shows a *possible* cause (but not yet a proof).
>
> 2) **Fig. 3-c**: survivor patients demonstrate a positive slope in the value of selected treatments after a flag is raised. In other words, their condition became high-risk, yet the treatment process did a good job, and the patients recovered. In direct contrast nonsurvivor patients show a steep downward trend following an inflection point, which indicates fast deterioration of patients’ condition.
>
> 3) Importantly, note that the SOFA score of both groups prior to the first raised flag is very similar. This rules out the argument where nonsurvivors’ condition gets worse simply because they were already in a bad condition. Indeed, the majority of all patients start their trajectory in more or less similarly bad conditions (otherwise they probably would not have been admitted to the ICU anyways). Furthermore, most nonsurvivors do not raise any flag in their first 8 hours. This fact rules out the argument that nonsurvivors, on the other hand, are deteriorating due to their own condition from beginning to the end. Hence, we can conclude that, at least on average, the selected treatments would play a role in the decline of patients’ condition.
>
>
> ``Comments on ethics:``
>
> We thank the reviewer for bringing this up. We do not believe that this work, or derivatives of it, should ever be used to exclude patients from treatment, e.g., not admitting patients based on dead-end analysis. We regret that our writing did not express this clearly enough. The treatment-avoidance part of our proposed approach is meant to shrink the scope of possible treatment options, and assist doctors in making better decisions. The potential for misuse of this model, and others like it, is legitimate and we have included a section on the ethical aspects and possible impacts of this work, copied here:
>
> >This work, or derivatives of it, should never be used to exclude patients from being treated, e.g., not admitting patients based on dead-end analysis. The treatment-avoidance part of our proposed approach is meant to shrink the scope of possible treatment options, and help the doctors to make better decisions. The flags that our approach, DeD, supplies are directly tied to specific treatment options rather than a binary treat/don’t treat decision. The intention of this approach is to assist clinicians by highlighting possibly unanticipated risks when making decisions and is not to be used as a stand-alone tool nor as a replacement of a human expert. Potential misuse of this algorithmic solution would carry significant risk to the well-being and survival of patients placed in a clinician’s care.
>
> >The primary goal of our work is to establish a proof of concept where especially high-risk treatments can be avoided, where possible, in context of a patient’s health condition. In acute care scenarios such as for patients with sepsis, treatments come with inherent risk profiles and potential harms. In these settings the tendency has become to overtreat patients with every attempt to ensure their survival, increasing the chance of clinical errors to occur [1]. Recent clinical research has sought to simplify practice to only the most necessary treatments [2]. In this spirit, we seek to assist clinicians by inferring the long-term impact of each available treatment in view of their risk of pushing the patient into an irrecoverable negative terminal condition.
>
> (Reference enumeration is in the context of the above paragraph)
>
> *[1] Carroll, Aaron E. "The high costs of unnecessary care." Jama 318.18 (2017): 1748-1749.*
>
> *[2] See: ​​https://www.bmj.com/too-much-medicine and http://jamanetwork.com/collection.aspx?categoryid=6017*

---

### Review · Ethics_Reviewer_7zjC · 2021-08-05

**Recommendation:**

I recommend the authors include a limitations and societal impact section which clearly outlines some of the issues, such as the "cost" of a false positive (however rare), recommendations for how such a model ought (not) to be used, etc.

**Ethical Issues:**

Yes

**Ethics Review:**

This paper discusses a technique for estimating which treatments are not going to help for sepsis patients and which patients have reached a medical dead-end-- there are no existing treatments that will lead to survival. The ethical issue comes in here in how such a model might be used. Could it be the case that the model would flag someone as in a dead-end who would in fact survive if treatment continued? It seems very unlikely given their low false positive rate, though generalization to other settings outside of which the data was collected is always an issue. Where does patient choice come into this? Given the high stakes involved in labeling someone as having no chance of survival, a discussion of how such a model might be used, what controls could be in place, and what the consequences of a bad prediction might be should be included.

---

> ### Author Response · Authors · 2021-08-16
> **Thank you, please review our responses to Reviewers 5pYT, AXB7 and Ethics Reviewer w9dW**
>
> We're grateful for the time you've spent to consider and review our work. We regret that we did not see nor were alerted to your review until after we submitted our responses to our original set of reviews. It appears that your review was completed before this time. We apologize for the oversight on our part. We have however, in the interim, addressed the ethical concerns raised in the original reviews and also included a new section that will be integrated in the main text of our paper.
>
> Included below is our response to Ethics Reviewer w9dW. We are grateful for the encouragement and suggestions to be more explicit about the potential limitations and ethical considerations of our proposed contributions. We regret that our paper didn't fully and clearly relay our thoughts and approach. We're grateful for the opportunity to improve our writing and the expected publication of our paper to better address ethical and societal implications of our work
>
> ===========
>
> Regarding the two cases of high-risk treatments vs. high-risk states, we agree with the reviewer that the distinction was not clear in the newly added section on ethics (*revised from our response to our original reviews and included below*). In short, we would like to highlight two points: A) DeD does both, and B) in both cases, DeD (due to approximation errors and false positive cases, even if the rate is very small) should not be used in isolation as a systematic way of say preventing patients from being admitted, or blindly ignore treatments. Rather, DeD is a source of technical advice based on AI methods, which implies possible errors and biases.
>
> As a methodological paper, the techniques and formal results presented in the paper could apply to many safety-critical sequential decision-making scenarios with limited offline data. In that regard, this paper provides a novel and effective way of identifying and diminishing the risk, which we firmly believe is useful. We acknowledge that this paper's method targets high-risk domains, which will involve ethical implications (as in healthcare). As such, we have added a section to directly clarify this and warn against possible misuse of our technique.
>
> We also highlight that the false positive rate is very small (further explained in the paper) and applying proper thresholds as a statistical step after neural net training can further significantly decrease false positive cases. In a healthcare setting, clinicians could change the thresholds (even at runtime) to make DeD less sensitive and further suppress identification of positive cases.
>
> Finally, as re-stated in our new ethics section, the proposed method is meant to serve as a warning system to help better calibrate human decisions, not a replacement. In the lack of large data containing sufficient exploratory alternative choices of actions and coverage of state, no methodology can become arbitrarily accurate. Hence, our technique may be crucial if we would like to utilize the *available* data to help the clinicians in most acute situations. Necessarily, this benefit must come with the knowledge of possible errors/biases and the acknowledgement of possible misconduct, which we, as scientists, would spell out in the paper.
>
> Here is the updated new section considering inputs from the ethics review and the above point:
>
> > This work, or derivatives of it, should never be used in isolation to exclude patients from being treated, e.g., not admitting patients based on dead-end analysis or blindly ignore treatments. The treatment-avoidance part of our proposed approach is meant to shrink the scope of possible treatment options, and help the doctors to make better decisions. One the other hand, signalling high-risk states is also meant to warn the clinicians for immediate attention before it becomes too late. In both cases, the flags that our approach, DeD, supplies are statistically tied to the training data and unavoidable sources of error and bias and should not be seen as a binary treat/don’t treat decision. In particular, even in the case of red flags, the signals should not be interpreted as *mathematical* dead-ends with full precision. The intention of our approach is to assist clinicians by highlighting possibly unanticipated risks when making decisions and is not to be used as a stand-alone tool nor as a replacement of a human expert. Potential misuse of this algorithmic solution would carry significant risk to the well-being and survival of patients placed in a clinician’s care.
>
> > The primary goal of our work is to establish a proof of concept where especially high-risk treatments can be avoided, where possible, in context of a patient’s health condition. In acute care scenarios such as for patients with sepsis, treatments come with inherent risk profiles and potential harms. In these settings the tendency has become to overtreat patients with every attempt to ensure their survival, increasing the chance of clinical errors to occur [1]. Recent clinical research has sought to simplify practice to only the most necessary treatments [2]. In this spirit, we seek to assist clinicians by inferring the long-term impact of each available treatment in view of their risk of pushing the patient into an irrecoverable negative terminal condition. The secondary goal of our work, on the other hand, is to call for emergency when the patient’s condition deteriorates, but is not necessarily realizable (yet) by the clinicians through monitoring clinical measures. This follows from the fact that DeD uses value functions, which provably enable such predictions.
>
> *[1] Carroll, Aaron E. "The high costs of unnecessary care." Jama 318.18 (2017): 1748-1749.*
>
> *[2] See: ​​https://www.bmj.com/too-much-medicine and http://jamanetwork.com/collection.aspx?categoryid=6017*

---

### Review · Ethics_Reviewer_w9dW · 2021-08-10

**Recommendation:**

In section 4, the authors write “Our goal is to identify all medical dead-end states, defined as patient states from which death is unavoidable, regardless of future treatments. Relatedly, we also desire to discover all treatments that may possibly lead to a medical dead-end state in order to learn which treatments to avoid”. Thus, when considering uses and ethical implications, I think it is helpful to differentiate between two potential uses of the methodology (1) identifying when the *current* state is a dead-end, (2) preventing *future* dead-ends.

In the newly added ethical implications section, the authors seem to suggest that their methodology should never be used for (1). This is not clear throughout the paper, however, e.g. the abstract says "We focus on patient condition in the intensive care unit, where a “medical dead-end indicates that a patient will expire, regardless of all potential future treatment sequences". Here, it is not clear before having read the paper that the authors are referring to a *future* dead-end; I encourage the authors to make the clarification early on, if this is the position they are taking.

Having said the above, choosing to withhold treatments that may cause harm when clinicians believe that the treatment is very unlikely to be effective for a patient is already part of clinical practice (e.g. withdrawing life-sustaining therapies of comatose patients who are deemed extremely unlikely to have a positive neurological recovery). Therefore, if the authors believe that their methodology should not be used to inform these type of decisions, it should be clear why. Is the risk of false dead-ends too high? Is it OK to use the proposed methodology to compare treatments relative to each other, but not to assign "dead-end" labels to patients? If so, why? It should be made clear what are the limitations of the methodology that deem it unfit for this task.

If the authors do believe that the methodology or its derivatives can (eventually) be used to identify current dead-ends and use this information to withhold treatments that are likely to cause harms and very unlikely to be effective, my recommendation would be: (1) make this explicit and clearly acknowledge the ethical risks, (2) connect this to bioethics literature and discussions on withholding treatment when it is deemed to be in the best interest of the patient, (3) clearly state new ethical considerations that arise from using this type of approach instead of current practices (e.g. the use of machine learning often introduces the risk of algorithmic bias, what are the open questions concerning potential bias for this methodology?).

To summarize, I believe the authors need to make a choice on whether they think that the methodology is appropriate for identifying when a patients' current state is a dead-end, which could subsequently inform the decision to withhold treatment. If the answer is no, make this clear throughout the paper and explain why the methodology is unfit for this task. If the answer is yes, update the newly added ethical discussion: acknowledge risks, connect to ethics discussions on the topic, and identify if new ethical considerations arise given the type of methodology and the risks associated to it. Naturally, the answer may be "not yet", if the authors think that the methodology is not inherently unfit for the task but there are important unanswered questions that should be addressed before, in which case potential harms or risks that are not yet studied or understood should be clearly stated.

**Ethical Issues:**

Yes

**Ethics Review:**

I agree with the reviewers who have flagged the ethical considerations. One of the possible uses of this methodology is to inform decisions to withhold treatment from a patient, which raises important ethical questions and risks.

I do not believe the ethical considerations should preclude publication of this work, but I do think it is something that the authors must explicitly address in the paper. I commend the authors for making a first step in this direction in the rebuttal. I still have important questions and suggestions that I elaborate on below.

---

> ### Author Response · Authors · 2021-08-16
> **Thank you! We appreciate the thoughts and suggestions.**
>
> We thank the reviewer for the detailed comments on the ethical implications of our paper.
>
> Regarding the two cases of high-risk treatments vs. high-risk states, we agree with the reviewer that the distinction was not clear in the newly added section on ethics (*revised from our response to our original reviews and included below*). In short, we would like to highlight two points: A) DeD does both, and B) in both cases, DeD (due to approximation errors and false positive cases, even if the rate is very small) should not be used in isolation as a systematic way of say preventing patients from being admitted, or blindly ignore treatments. Rather, DeD is a source of technical advice based on AI methods, which implies possible errors and biases.
>
> As a methodological paper, the techniques and formal results presented in the paper could apply to many safety-critical sequential decision-making scenarios with limited offline data. In that regard, this paper provides a novel and effective way of identifying and diminishing the risk, which we firmly believe is useful. We acknowledge that this paper's method targets high-risk domains, which will involve ethical implications (as in healthcare). As such, we have added a section to directly clarify this and warn against possible misuse of our technique.
>
> We also highlight that the false positive rate is very small (further explained in the paper) and applying proper thresholds as a statistical step after neural net training can further significantly decrease false positive cases. In a healthcare setting, clinicians could change the thresholds (even at runtime) to make DeD less sensitive and further suppress identification of positive cases.
>
> Finally, as re-stated in our new ethics section, the proposed method is meant to serve as a warning system to help better calibrate human decisions, not a replacement. In the lack of large data containing sufficient exploratory alternative choices of actions and coverage of state, no methodology can become arbitrarily accurate. Hence, our technique may be crucial if we would like to utilize the *available* data to help the clinicians in most acute situations. Necessarily, this benefit must come with the knowledge of possible errors/biases and the acknowledgement of possible misconduct, which we, as scientists, would spell out in the paper.
>
> Here is the updated new section considering inputs from the ethics review and the above point:
>
> > This work, or derivatives of it, should never be used in isolation to exclude patients from being treated, e.g., not admitting patients based on dead-end analysis or blindly ignore treatments. The treatment-avoidance part of our proposed approach is meant to shrink the scope of possible treatment options, and help the doctors to make better decisions. One the other hand, signalling high-risk states is also meant to warn the clinicians for immediate attention before it becomes too late. In both cases, the flags that our approach, DeD, supplies are statistically tied to the training data and unavoidable sources of error and bias and should not be seen as a binary treat/don’t treat decision. In particular, even in the case of red flags, the signals should not be interpreted as *mathematical* dead-ends with full precision. The intention of our approach is to assist clinicians by highlighting possibly unanticipated risks when making decisions and is not to be used as a stand-alone tool nor as a replacement of a human expert. Potential misuse of this algorithmic solution would carry significant risk to the well-being and survival of patients placed in a clinician’s care.
>
> > The primary goal of our work is to establish a proof of concept where especially high-risk treatments can be avoided, where possible, in context of a patient’s health condition. In acute care scenarios such as for patients with sepsis, treatments come with inherent risk profiles and potential harms. In these settings the tendency has become to overtreat patients with every attempt to ensure their survival, increasing the chance of clinical errors to occur [1]. Recent clinical research has sought to simplify practice to only the most necessary treatments [2]. In this spirit, we seek to assist clinicians by inferring the long-term impact of each available treatment in view of their risk of pushing the patient into an irrecoverable negative terminal condition. The secondary goal of our work, on the other hand, is to call for emergency when the patient’s condition deteriorates, but is not necessarily realizable (yet) by the clinicians through monitoring clinical measures. This follows from the fact that DeD uses value functions, which provably enable such predictions.
>
> *[1] Carroll, Aaron E. "The high costs of unnecessary care." Jama 318.18 (2017): 1748-1749.*
>
> *[2] See: ​​https://www.bmj.com/too-much-medicine and http://jamanetwork.com/collection.aspx?categoryid=6017*

---

### Decision · Program_Chairs · 2021-09-27

**Decision:**

Accept (Poster)

**Comment:**

The reviewers appreciated the paper and agree it provides an interesting new problem and a useful new method. The authors are expected to address the points raised by reviewers in a final version, including as they outlined in their response. Most crucially, the authors should address the issues raised by the ethics reviewers: the additional discussion outlined by the authors in their response was deemed an appropriate way to do this and the authors need to implement this.